

# Large-scale effects of benthic fauna on carbon, nitrogen, and phosphorus dynamics in the Baltic Sea

Eva Ehrnsten[1], Oleg Pavlovitch Savchuk[1], Bo Gustav Gustafsson[1,2]

[1]Baltic Nest Institute, Baltic Sea Centre, Stockholm University, Stockholm, 10691, Sweden
[2]Tvärminne Zoological Station, University of Helsinki, Hanko, 10900, Finland

*Correspondence to*: Eva Ehrnsten (eva.ehrnsten@su.se)

**Abstract.** Even though the effects of benthic fauna on aquatic biogeochemistry have been long recognized, few studies have addressed the combined effects of animal bioturbation and metabolism on ecosystem–level carbon and nutrient dynamics. Here we merge a model of benthic fauna (BMM) into a physical–biogeochemical ecosystem model (BALTSEM) to study the

long–term and large–scale effects of benthic fauna on nutrient and carbon cycling in the Baltic Sea. We include both the direct effects of faunal growth and metabolism and the indirect effects of its bioturbating activities on biogeochemical fluxes of and transformations between organic and inorganic forms of carbon (C), nitrogen (N), phosphorus (P) and oxygen (O). Analyses of simulation results from the Baltic Proper and Gulf of Riga indicate that benthic fauna makes up a small portion of seafloor organic stocks, but contributes considerably to benthic–pelagic fluxes of inorganic C, N and P through its metabolism. Results

also suggest that the relative contribution of fauna to mineralisation of sediment organic matter increases with increasing nutrient loads. Further, bioturbation decreases benthic denitrification and increases P retention in sediments, the latter having far–reaching consequences throughout the ecosystem. Reduced benthic–pelagic P fluxes lead to a reduction of N fixation and primary production, lower organic matter sedimentation fluxes and thereby generally lower benthic stocks and fluxes of C, N and P. This chain of indirect effects overrides the direct effects of faunal respiration, excretion and bioturbation. Due to large

uncertainties related to parameterization of benthic processes, we consider this modelling study a first step towards disentangling the complex large–scale effects of benthic fauna on biogeochemical cycling.

**Keywords:** benthic fauna, metabolism, bioturbation, carbon, nitrogen, phosphorus, physical–biogeochemical modelling, benthic-pelagic coupling



## 1 Introduction

Coastal ecosystems are highly productive, consist of diverse biological communities and carry out important functions including those supporting a growing world population (Costanza et al., 1997, 2014). However, they are facing multiple anthropogenic pressures such as nutrient loading and climate change (Cloern et al., 2016; Halpern et al., 2008). Elucidating the mechanisms of the coupled biogeochemical cycling of carbon (C), nitrogen (N) and phosphorous (P) in these systems is

important to understand how they respond to current and future pressures, but also because they contribute to the regulation of global climate and nutrient cycles by processing anthropogenic emissions from land before they reach the ocean (Ramesh et al., 2015; Regnier et al., 2013a, 2013b; Seitzinger, 1988).

In contrast to the deep open ocean, benthic-pelagic coupling plays a large role in biogeochemical cycling in coastal and estuarine ecosystems (Soetaert and Middelburg, 2009). Coastal sediments act as hotspots for organic matter degradation and

permanent removal of elements from biological cycling through burial and denitrification (Asmala et al., 2017; Regnier et al., 2013a; Seitzinger, 1988). The bioturbating activities of benthic fauna alter the physical and chemical properties of surface sediments, which in turn strongly influence organic matter degradation processes and benthic–pelagic biogeochemical fluxes (Aller, 1982; Rhoads, 1974; Stief, 2013). Additionally, benthic fauna retain carbon and nutrients in its biomass and transform them between organic and inorganic forms through metabolic processes (Ehrnsten et al., 2020b and references therein; Herman

et al., 1999; Josefson and Rasmussen, 2000). Together, these direct and indirect effects of benthic fauna have far–reaching consequences for ecosystem functioning in the benthic and pelagic realms (Griffiths et al., 2017; Lohrer et al., 2004).

Even though the importance of benthic fauna for sediment biogeochemistry and benthic–pelagic fluxes has long been recognized (Rhoads, 1974), the combined effects of animal bioturbation and metabolism have seldom been studied together (Ehrnsten et al., 2020b; Middelburg, 2018; Snelgrove et al., 2018). Further, empirical studies of faunal effects often focus on

temporally and spatially limited parts of the system, omitting important interactions and variability occurring in natural ecosystems (Snelgrove et al., 2014). Here, we extend a physical–biogeochemical model of the Baltic Sea ecosystem (BALTSEM; Gustafsson et al., 2014; Savchuk et al., 2012) with benthic fauna components based on the Benthic Macrofauna Model (BMM; Ehrnsten et al., 2020a). We include both the direct feedbacks from animal growth and metabolism and the indirect effects of their bioturbating activities on biogeochemical cycling to evaluate their relative contributions.

We use the Baltic Sea as a model area for two reasons: (i) the shallow depth (mean depth 57 m) and enclosed geography with a long water residence time (about 33 years) contribute to strong benthic-pelagic coupling (Snoeijs-Leijonmalm et al., 2017; Stigebrandt and Gustafsson, 2003), and (ii) the major features of biogeochemical cycling of C, N and P in the Baltic Sea are well known due to a wealth of oceanographic measurements and studies performed over the past century, making it an ideal system for process-based modelling (Eilola et al., 2011; Gustafsson et al., 2017; Savchuk and Wulff, 2009, 2001). However,

the sediment pools and the role of sediment processes in benthic–pelagic exchange are not as well quantified as pelagic pools





and fluxes. Here, we aim to fill this knowledge gap and explore the role of benthic fauna in biogeochemical cycling of C, N and P on a long–term ecosystem–level scale.

## 2 Materials and methods

### 2.1 Study system

The Baltic Sea is a semi–enclosed coastal sea in northern Europe with strong latitudinal and depth gradients in salinity, temperature and productivity shaping the distribution of species and ecosystem functioning (Bonsdorff, 2006; Elmgren, 1984; Snoeijs-Leijonmalm et al., 2017). The diversity of benthic fauna is low due to the low salinity, and large, deep-burrowing species are only found near the entrance to the Baltic Sea (Bonsdorff, 2006; Remane, 1934). Thus, the sediment layer mixed by bioturbating animals is very shallow compared to other coastal and shelf seas (Teal et al., 2008), and sediment oxygen

penetration depth is usually counted in millimetres rather than centimetres (Almroth-Rosell et al., 2015; Bonaglia et al., 2019; Hermans et al., 2019). Nonetheless, several studies have measured significant effects of benthic fauna on benthic nutrient processing in the Baltic Sea (e.g. Berezina et al., 2019; Lehtonen, 1995; Norkko et al., 2013, 2015).

 Due to its large catchment area and limited water exchange with the North Sea, the Baltic Sea is heavily influenced by anthropogenic nutrient emissions (Andersen et al., 2017; Gustafsson et al., 2012). Although emissions have been significantly

reduced since the peak in the 1980s, recovery from eutrophication is slow with limited reductions in nutrient pools and primary productivity seen to date (Gustafsson et al., 2012; Savchuk, 2018; Zdun et al., 2021). This is due to the long water residence time and the build–up of nutrient stores in soils and marine sediments during several decades (McCrackin et al., 2018; Savchuk, 2018 and references therein).

In this study, we focus on the Baltic Proper and the Gulf of Riga (Fig. 1). The Baltic Proper is the central, deepest basin of the

Baltic Sea with a maximum depth of 459 m and a mean depth of ca 75 m. A permanent halocline at ca 60–80 m limits the vertical mixing between the low–salinity surface waters (5–8 psu) and the deeper waters with a salinity of 9–13 (Snoeijs-Leijonmalm et al., 2017). A majority of the waters below the halocline are hypoxic or anoxic because the mineralization of organic matter sinking through the water column and in the sediments consumes oxygen faster than it is replenished by infrequent salt water intrusions and vertical turbulent mixing. The expanding hypoxia has severely reduced the area habitable

by benthic fauna in the Baltic Sea (Carstensen et al., 2014a, 2014b). In the reducing environment, P bound to iron-humic complexes is released from sediments and contribute to the dissolved inorganic P (DIP) pool in the water column. The excess DIP promotes the fixation of atmospheric N by cyanobacteria, in turn promoting primary production by other phytoplankton, which leads to increased sinking and mineralization of organic matter, in turn expanding hypoxia. This feedback loop, termed the 'vicious circle' (Vahtera et al., 2007) is further strengthened by climate change, as increasing water temperatures promote

cyanobacterial blooms (Kahru et al., 2020; Kahru and Elmgren, 2014).

The Gulf of Riga is a semi-enclosed coastal bay with mean and maximum depths of 23 and 51 m, respectively, and a salinity of 4–7 (Snoeijs-Leijonmalm et al., 2017). In contrast to the Baltic Proper, the Gulf of Riga is relatively well mixed and hypoxia only occurs intermittently under the summer thermocline (Kotta et al., 2008), accompanied by increased release of phosphate from the sediments (Eglite et al., 2014). When occurring more often in the recent decade, the intensity and extension of both sporadic hypoxia and phosphate release have somewhat increased (HELCOM, 2018; Stoicescu et al., 2021). Similarly, the summer cyanobacteria blooms sporadically occurring in the Gulf of Riga before the 2010s (Kahru and Elmgren, 2014), regularly and extensively cover the gulf since 2015 (pers. comm. Mati Kahru).

## 2.2 Model description

The biogeochemical cycling in the BALTSEM model was extended to include benthic fauna. BALTSEM simulates physical circulation and biogeochemical transformations of C, N, P, O and Si in the Baltic Sea in response to climatic conditions and nutrient inputs from rivers, point sources and atmospheric deposition. It describes the Baltic Sea as 13 horizontally homogenous boxes with a dynamic depth resolution of generally less than 1 m in the pelagial (Fig. 1). Sediments are represented as terraces at 1 m depth intervals with an area corresponding to the hypsography of each basin. The new benthic components were constructed from the carbon-based Benthic Macrofauna Model (BMM) described in Ehrnsten et al. (2019b, 2019a, 2020a) extended to include nitrogen (N) and phosphorus (P) components.

Below we give a short description of the benthic dynamics in the new model version, referred to as BALTSEM–BMM, with a focus on the effects of benthic fauna on biogeochemical processes (Fig. 2). A full mathematical description of the benthic biogeochemical processes of the model is found in Appendix A. For a description of the pelagic biogeochemistry and physics, we refer the reader to Gustafsson et al. (2012, 2014) and Savchuk et al. (2012). Additionally, all benthic and pelagic state variables are listed in Table A1.

### 2.2.1 Benthic fauna dynamics

BALTSEM–BMM includes C, N, and P contents in biomass of three functional groups of benthic fauna. The facultative deposit–/suspension–feeding bivalve *Limecola balthica* is a key species dominating the biomass of benthic communities in large parts of the sea and is therefore represented by its own state variable. The group "deposit–feeders" represents the amphipods *Monoporeia affinis* and *Ponotporeia femorata,* the invasive polychaetes *Marenzelleria* spp. and other macrofaunal species dependent on surface sediment organic matter as their primary food source. The group "predators" represents species feeding on the two former groups, such as the isopod *Saduria entomon,* the polychaete *Bylgides sarsi* and the priapulid *Halicryptus spinulosus.*

The biomass of all groups of fauna are modelled as a dynamic mass balance between fluxes formed by food uptake, assimilation, respiration or excretion, and mortality. The formulations for dynamics of the functional groups and their food banks were kept as in BMM (Ehrnsten et al., 2019a, 2020a) as far as possible. The main change is the addition of N and P





components to each state variable. Consumers generally regulate their inner stoichiometry within tight limits (Sterner and Elser, 2002), therefore the fauna was given a constant C:N:P ratio. The ratio was approximated based on measured ratios for the dominating species in the Baltic Sea (see Appendix A).

Food uptake is modelled as a function of food availability in C units. The uptake of N and P components of a food source are thereafter calculated proportionally to the C:N:P ratio of the food source. Part of the food is assimilated, with an assimilation factor depending on the food source. The assimilation factors for C components were applied to N and P as well (Table A3). The unassimilated part is released as faeces, adding organic matter to the sediment C, N and P pools (Fig. 2).

Respiration and excretion of inorganic C, N and P is divided into three parts: (1) a basal maintenance part related to biomass;
(2) a growth and activity part related to food uptake as a proxy for activity; and (3) excess excretion. As the stoichiometry of assimilated food varies, excretion of excess elements is calculated dynamically to keep the fixed stoichiometry of the benthos. Formulations are similar to those used for zooplankton in BALTSEM and for benthos in other ecosystem models (Ebenhöh et al., 1995; Spillman et al., 2008). Respiration and excretion fluxes add to the bottom water pools of dissolved inorganic carbon (DIC), nitrogen (NH, representing total ammonia) and phosphorus (PO, representing total phosphate). Respiration also
consumes bottom water oxygen with a respiratory quotient of 1 mol $O_2$: mol $CO_2$ (Brey, 2001).

### 2.2.2 Sediment dynamics and bioturbation

As in the standard BALTSEM, sediment bioavailable C, N, P and Si are represented as vertically integrated concentrations in the biogeochemically active surface layer of unspecified thickness. The concentrations are modelled as a dynamic mass balance between fluxes formed by sedimentation, degradation and burial, extended by interactions with benthic fauna. Sediment C, N
and P pools are further divided into three banks of different age to resolve the food limitation of benthic fauna (Fig. 2), while benthic Si is represented as a single pool. Oxygen is not a state variable in sediments, but several benthic processes interact with simulated bottom water oxygen.

Bioturbation by benthic fauna, including sediment reworking and burrow ventilation, generally increases the oxygenation of sediments (Michaud et al., 2005; Volkenborn et al., 2012), promoting the binding of phosphate to iron oxides (P sequestration)
and stimulating nitrogen oxidation (Ekeroth et al., 2016; Norkko et al., 2012; Renz and Forster, 2014). Similar to Isaev et al. (2017), we use simple formulations for the effects of faunal activities on the oxygen-dependent processes of sediment nitrification/denitrification and P sequestration through a bioturbation enhancement factor $E_{bio}$. The formulation for $E_{bio}$ was taken from Blackford (1997), using the feeding rate of fauna as a proxy of its bioturbation activity:

$$E_{bio} = E_{max} \left( \frac{\sum_{i=1}^{3}(cf_{BFi}U_{BFiC})}{\sum_{i=1}^{3}(cf_{BFi}U_{Ci}) + K_{bio}} \right) \tag{1}$$





where $E_{max}$ is the maximum enhancement, $cf_{BFi}$ is a contribution factor of functional group $i$. $U_{BFiC}$ is the carbon uptake rate of
group $i$ and $K_{bio}$ is a half saturation constant. As *L. balthica* is more sedentary than the other groups, a contribution factor of
0.5 was assigned to it and a factor of 1 to the two other groups (Ebenhöh et al., 1995; Gogina et al., 2017 and refences therein).

Within each of the three sediment banks, C, N and P components share the same source and sink processes. Sinking organic
matter is integrated into a bank of fresh organic matter available as food for deposit–feeders. This bank ages into a slightly
older bank available as food for *L. balthica* only. The second bank ages into a third bank considered unavailable as food for
benthic fauna, but available for bacterial mineralization.

For each element, mineralization fluxes from the three sediment banks are combined into a total flux ($Z_{SEDXtot}$, $X$ = C, N, P).
For C, the total sediment mineralization flux directly adds to the pelagic DIC pool. N and P mineralization fluxes are further
divided in the same way as in the standard BALTSEM (Gustafsson et al., 2014; Savchuk, 2002) with the addition of
bioturbation effects (Fig. 2b–c, Eq. 1–9).

Depending on oxygen concentrations in the bottom water layer and bioturbation intensity, mineralized sediment N is released
to the water column as ammonia ($O_{NH}$, Eq. 2) or oxidised N ($O_{NO}$, NO representing $NO_2$ and $NO_3$, Eq. 3) or denitrified to $N_2$
($W_{Deni}$, Eq. 4).

$$O_{NH} = v_{NOXY} Z_{SEDNtot} \qquad (2)$$

$$O_{NO} = (1 - v_{NOXY}) \eta_N Z_{SEDNtot} \qquad (3)$$

$$W_{Deni} = (1 - v_{NOXY})(1 - \eta_N) Z_{SEDNtot} \qquad (4)$$

In anoxic or nearly anoxic conditions, the majority of mineralized N is released to the water column as NH, defined by the
proportion $v_{NOXY}$ (Eq. 5), where *OXY* is bottom water total oxygen concentration (that is allowed to become negative to
represent hydrogen sulphide), and $a_{vNOXY}$ is a fitting constant. Otherwise the mineralized N is oxidized.

$$v_{NOXY} = \left(1 + \frac{(OXY + |OXY| + a_{vNOXY})}{(|OXY| + a_{vNOXY})^8}\right)^{-1} \qquad (5)$$

Subsequently, the oxidized portion can be denitrified into $N_2$ or released to the pelagic NO pool. Denitrification is treated as a
permanent sink for N. The proportion released as NO ($\eta_N$) is positively related to oxygen according to Eq. (7), where $q_N$, $a_N$,
$b_N$ and $c_N$ are fitting constants and $E_{bio}$ is the bioturbation enhancement factor.

$$\eta_N = q_N \left(\frac{1}{1 + exp\left(a_N - (b_N + \boldsymbol{E_{bio}})(c_N + max(OXY, 0))\right)} - \frac{1}{1 + exp(a_N - b_N c_N)}\right) \qquad (7)$$

A fraction $\eta_P$ of mineralized sediment P is sequestered in the sediment (Eq. 8), while the rest is released as phosphate ($O_{PO}$) to
the pelagic PO pool (Eq. 9).





$$K_P = \eta_P Z_{SEDPtot} \qquad (8)$$

$$O_{PO} = (1 - \eta_P)Z_{SEDPtot} \qquad (9)$$

The fraction sequestered is positively related to oxygen and bioturbation and negatively related to salinity according to Eq.
(10), where $q_P$, $a_P$, $b_P$, $c_P$, $d_P$, $e_P$, $f_P$ and $g_P$ are fitting constants. The fraction has an upper limit of 1 (100% of mineralized P sequestered), but can take on negative values, representing a release of previously sequestered P in severely hypoxic or anoxic conditions. The salinity dependence $f_{SAL}$ in the third term is used as a proxy for the higher availability of the phosphate-binding agents (e.g. iron and humic substances) in the fresher Gulf of Bothnia.

$$\eta_P = q_P \tanh(a_P \, OXY) + \frac{b_P(1+E_{bio})max(OXY,0)}{c_P + max(OXY,0)} - f_{SAL} \qquad (10)$$

**2.3 Simulations**

The model was run over 1970–2020 forced with observed nutrient loads and actual weather conditions as described in Gustafsson et al. (2012, 2017) with forcing time–series extended to 2020. Initial conditions in 1970 were based on observations for pelagic variables and hindcast simulations for benthic variables as described in Gustafsson et al. (2012) and Ehrnsten et al. (2020a). As the purpose of this study was to evaluate large–scale dynamics, results were aggregated as means and standard deviations of the last two decades (2000–2020) to capture differences in long-term averages while accounting for interannual variations.

It is difficult to constrain the new parameters as well as to validate the large–scale dynamics against observations from the field or laboratory, which are usually made on much smaller temporal and spatial scales. Instead, we made a sensitivity analysis testing the effects of changing the parameter $E_{max}$ in the range 0 to 0.6, where 0 represents no bioturbation and 0.6 is the theoretical maximum value of $E_{bio}$, giving 100% P sequestration. $E_{max} = 0.3$ was used in the default model run.

Additionally, we estimate the contribution of bioturbation to benthic–pelagic nutrient fluxes by calculating the theoretical fluxes without bioturbation enhancement (i.e. with $E_{bio} = 0$) for each time–step, while running the default model with bioturbation. In contrast to the sensitivity analysis, this analysis shows the direct effects of bioturbation without accounting for indirect effects through the ecosystem.

Finally, to study the relationship between nutrient loads and the role of benthic fauna in biogeochemical cycling, we ran two future scenarios 2021-2100 with either decreasing loads of N and P according to the Baltic Sea Action Plan (BSAP scenario, total loads to the Baltic Sea 739 kton N year[-1] and 21 kton P year[-1]) or increasing loads corresponding to the highest recorded historical loads based on mean N and P loads of 1980-1990 (HIGH load scenario, 1235 kton N year[-1] and 69 kton P year[-1]). For comparison, the average nutrient loads in 2000-2020 were 936 kton N year[-1] and 36 kton P year[-1]. The scenarios were combined with a statistical climate forcing representing no change in climate. Details of the scenarios can be found in Ehrnsten et al. (2020). These results were also aggregated over the last two decades (2080-2100).




## 3 Results

### 3.1 Validation

The BALTSEM–BMM behaves very similarly to the standard BALTSEM model, which has been extensively validated (Gustafsson et al., 2012, 2014; Savchuk et al., 2012) and shown to perform favourably in relation to similar Baltic Sea models (Eilola et al., 2011; Meier et al., 2018). A comparison of the main pelagic state variables (salinity, temperature and concentrations of oxygen, NH, NO, and PO) to observations over time (1970–2015) and depth shows an overall relative bias of 1.40, while the relative bias of the standard BALTSEM is 1.41. The relative bias index compares model-data difference with variability in the data, giving an estimate of how well the model captures variability in nature on seasonal, annual and
decadal scales (Savchuk et al., 2012). A detailed description and results of this analysis are found in Appendix B.

Ehrnsten et al. (2020) did a comprehensive validation of simulated biomasses of benthic fauna against observations over depth intervals in the largest basins of the Baltic Sea. We re-ran this analysis with the results of the coupled model and extended it to include the southern and southwestern basins. The extended analysis, based on 7679 observations, confirms previous results that the model captures the main observed patterns of biomass over latitude and depth with reasonable accuracy (Fig. 3,
Appendix C). Simulated mean biomasses of the individual functional groups and the groups combined were mostly within one standard deviation of observed means from the Arkona Basin (basin 7) in the south to the Bothnian Bay (basin 11) in the north, although it should be noted that the spread of observed data is large. In the high–salinity Kattegat and Danish Straits near the entrance to Baltic Sea, the model is not applicable as the benthic biomass is dominated by groups not included in the present model, such as suspension-feeding bivalves and large echinoderms. Further details on this analysis are presented in Appendix
C.

The databases used did not contain observations of benthic fauna from the Gulf of Riga, therefore we compared the simulated biomasses to estimates from the literature (Table 1). The simulated biomass of benthic fauna in the gulf varied substantially over depth and time (29–284 g wwt m$^{-2}$), which is supported by field observations. The simulated mean biomass is within the large range of estimates from the literature.

Carman and Cederwall (2001) have estimated the amounts of C, N and P in Baltic Sea sediments based on core samples. It is not straightforward to compare the total amounts to simulations, as the thickness of the simulated sediment is not defined. However, the estimated C:N:P ratios can be more readily compared. In the Baltic Proper, the molar C:N:P ratio (calculated from their Table 11.4) was estimated to be 116:12:1 in the top centimetre of sediments and 137:14:1 in the top 5 cm, while the simulated ratio was 108:15:1. In the Gulf of Riga, the estimated C:N:P ratio was 73:7:1 in the top 1 cm and 83:8:1 in the top
5 cm, while the simulated ratio was 75:10:1.

### 3.2 Budgets of benthic C, N and P

Long–term (2000–2020) average benthic budgets of C, N and P are shown in Fig. 4 for the Baltic Proper and Gulf of Riga. The results for the Baltic Proper are restricted to the depth interval 0–90 m, as benthic fauna is practically absent in the oxygen–poor waters below this depth (Fig. 5a). Figure 5 also shows the long–term average depth distribution of the bioturbation factor $E_{bio}$ in the two basins.

According to the default simulation, the benthic fauna made up a minor part of the benthic organic C, N and P stocks (1–4 %), but had a proportionally larger share in benthic–pelagic fluxes of DIC (23 % and 31 % in the Baltic Proper and Gulf of Riga, respectively), DIN (43 % and 51 %), and DIP (25 % and 34 %). The budgets also show that input of organic matter to the sediments was higher in the Gulf of Riga compared to the Baltic Proper, resulting in overall higher benthic stocks and benthic–pelagic fluxes (Figs. 4 and 5).

### 3.3 Bioturbation effects on C, N and P dynamics

When accounting only for the direct effects of bioturbation, it increased NO outflux by 0.41 g N m$^{-2}$ year$^{-1}$ (+40 %) and decreased sediment denitrification by the same amount (-14 %) in the Baltic Proper (simulated average of 2000–2020, Fig. 6a). Similarly, P sequestration was increased and PO outflux decreased by 0.09 g P m$^{-2}$ year$^{-1}$ (+31 % and -15 %), respectively. In the Gulf of Riga, both the absolute flux rates and the relative effects of bioturbation on them were larger than in the Baltic Proper (Fig 6b).

When also accounting for the indirect effects of bioturbation in the sensitivity analysis, changes were more complex (Figs. 7–8). For example, comparing the default run ($E_{max}$ = 0.3) to the run with no bioturbation ($E_{max}$ = 0) in the Baltic Proper, denitrification was reduced by 0.44 g N m$^{-2}$ year$^{-1}$ but NO outflux increased by only 0.38 g N m$^{-2}$ year$^{-1}$ (Fig 7c), while P sequestration increased by 0.14 g P m$^{-2}$ year$^{-1}$ and PO outflux decreased by 0.03 g P m$^{-2}$ year$^{-1}$ (Fig. 7e).

In general, increasing bioturbation led to a decrease of most benthic stocks (Fig. 9) and fluxes (Fig 7). This can be explained by the following chain of effects. Increased P sequestration in the sediments (Fig. 7e–f) led to less pelagic DIP available for phytoplankton, especially cyanobacteria, growth and thereby lower N fixation and primary production (Fig. 8) which in turn led to lower organic matter sedimentation rates, lower sediment stocks of organic matter and consequently lower rates of most sediment biogeochemical transformations and fluxes (Figs. 7–9). Decreasing organic matter sedimentation also led to decreased biomass of benthic fauna (primarily due to a reduction in *L. balthica* biomass) and excretion of DIN and DIP. An exception is the sediment P stock (Fig 9c), which increased with bioturbation despite decreased sedimentation of organic P (Fig. 7f). This was due to the increased sequestration adding to the 3$^{rd}$ sediment P bank.

### 3.4 Nutrient load scenarios

All results below are calculated from means of 2080-2100 for the BSAP and HIGH nutrient load scenarios and 2000-2020 for the default model run in the Baltic Proper (0-90 m depth) and Gulf of Riga.

With increasing nutrient loads, primary production and input of particulate organic matter (POM) to the sediments increased (Fig. 10a–c), resulting in an increase of most benthic stocks and fluxes. The biomass of benthic fauna responded more strongly to changing nutrient loads than the bioturbation enhancement coefficient linked to the feeding activities of fauna (Fig. 10d–e).

With changing loads, the relative roles of faunal and microbial processes in the sediment changed (Figure 11). With increasing loads, an increasing proportion of benthic-pelagic fluxes of inorganic nutrients originated from faunal metabolism. Expressed as percent of POM input to the sediments, the respiration and excretion of fauna was 12–13 % in the BSAP scenario and 23–24 % in the HIGH scenario in the Baltic Proper. In the Gulf of Riga, respiration and excretion was 23–24 % of POM input in the BSAP scenario and 35–37 % in the HIGH scenario. Correspondingly, the proportions of POM input released as dissolved

inorganic substances resulting from microbial processes in the sediment (DIC, NO+NH and PO outflux in Fig. 11) were lower in the BSAP than in the HIGH scenario.

The relative proportion of POP input sequestered shows a complex pattern in the Gulf of Riga (Fig. 11f): the proportion of POP input sequestered was lower in the BSAP scenario compared to the default model due to less fauna and thereby less bioturbation. However, the proportion was also lower in the HIGH load scenario, as the increased occurrence of hypoxia (Fig.

10f) counteracted the effects of increased bioturbation. In the Baltic Proper, relative P sequestration shows a decreasing pattern with increasing loads driven by increasing occurrence of hypoxia (Fig. 11e).

Even though the total amount of POM input to the sediment increased with increasing nutrient loads, it constituted a decreasing proportion of primary production. In the BSAP scenario, almost half of the annual primary production reached the seafloor (48 % and 47 % in the Gulf of Riga and Baltic Proper, respectively) compared to 21 % and 27 % in the HIGH load scenario.

Thus, the proportion of primary production mineralized by fauna varied only slightly with nutrient load scenario because of the opposite responses of sinking organic matter and fauna: 4.7 % (BSAP) to 4.1 % (HIGH) in the Baltic Proper and 10.7 % (HIGH) to 9.6 % (BSAP) in the Gulf of Riga.

## 4 Discussion

We have created a new tool to simulate the long–term and large–scale effects of benthic fauna on biogeochemical cycling in

a coastal sea by fully merging two existing process–based models. First simulations with the new model indicate that the benthic fauna makes up a small part of benthic organic stocks, but contributes substantially to organic matter mineralization and benthic–pelagic fluxes of inorganic C, N and P through its metabolism. Further, the stimulation of P binding in sediments by bioturbation significantly reduced N fixation and primary production in the simulations, indicating that benthic fauna can alleviate the 'vicious circle' of eutrophication.

**4.1 Model performance**

In general, the BALTSEM-BMM model reproduces the observed Baltic Sea–scale patterns of decreasing biomass of benthic fauna with latitude and depth reasonably well, as also shown for a previous one-way coupled model version (Ehrnsten et al.,

2020a). Compared to observations, the model seems to underestimate the biomass of benthic fauna in the Bothnian Sea and overestimate it in the Gulf of Finland. The former may be due to an underestimation of primary productivity in the Bothnian

Sea by BALTSEM, while the omission of possible negative effects of low salinity on *L. balthica* in the Gulf of Finland may explain the latter (see Appendix C). Additionally, the model would need an addition of several groups of "megabenthos" (e.g. large echinoderms and suspension-feeding bivalves) to be applicable to the marine areas at the entrance to the Baltic Sea.

The simulated mean biomass of benthic fauna in Gulf of Riga was more than twice as high as the recent estimate by Gogina et al. (2016). Possible reasons for overestimation may be that the model does not take into account the limitations by mobile

substrates and low salinity, especially in the southern part of the basin (Carman et al., 1996; Kotta et al., 2008). In this region, a reduction of benthic biomass (e.g. of *Monoporeia affinis* and *Limecola balthica*) occurred in the 1990s for reasons not well understood (Kortsch et al., 2021). On the other hand, our biomass estimate is less than half of that by Kotta et al. (2008), assuming reported dry weight is 10 % of wet weight.

The modelled patterns in total benthic biomass are strongly driven by changes in *L. balthica* biomass in response to changes

in food availability, leading to extinction of the group in deep waters and in the oligotrophic Bothnian Bay. These patterns are strongly supported by observations.

The BALTSEM model was neither improved nor worsened by the addition of benthic fauna, according to the performance analyses comparing pelagic variables to observations (Appendix B). This shows that increasing model complexity does not necessarily increase accuracy, especially when the functions and/or variables added are not well known (Ehrnsten et al., 2020b;

Levins, 1966). In general, though, the previous assessments of model performance showing that the model is able to reproduce seasonal and long-term variations in biogeochemical variables and performs well in comparison to other Baltic Sea models, remain valid (Eilola et al., 2011; Gustafsson et al., 2012, 2014; Meier et al., 2018; Savchuk et al., 2012).

Unfortunately, we cannot properly validate the simulated sediment stocks or fluxes due to a lack of large–scale data and insufficient understanding of the multitude of mechanisms underlying the biogeochemical transformations and fluxes. Thus,

the quantitative results should be viewed with caution. The main strength of this study is instead in the "what–if" analysis showing the many interlinkages among C, N, P and O cycles and between benthic and pelagic processes.

### 4.2 Biogeochemical effects of benthic fauna

Similar to estimates made with previous uncoupled versions of the model (Ehrnsten et al., 2019a, 2020a), the results of this study suggests that respiration by fauna constitutes a significant part of organic matter mineralization in sediments. The fauna

mineralized about 8–17 g C m$^{-2}$ year$^{-1}$ or 22–31 % of POC input to the sediments in 2000-2020. This agrees well with previous estimates from the Baltic Sea of 22–40 % (Ankar, 1977; Elmgren, 1984; Kuparinen et al., 1984). Similarly, Rodil et al. (2019) estimated that macrofauna contributed 18–26 % of total benthic respiration in soft and 11–45 % in hard bottom sites in the Gulf of Finland. Herman et al. (1999) estimated that respiration by macrofauna mineralizes 5–25 % of annual primary





production in shallow estuaries. In these deeper coastal areas, we estimate that the fauna mineralized 3–9 % of annual primary
production in the Baltic Proper and 8–15 % in the Gulf of Riga in 1970-2020, with considerable interannual variations.

The sensitivity analysis showed a large effect of bioturbation on primary production levels mainly due to increased P retention. When bioturbation increased P sequestration, this led to a weakening of the 'vicious cycle' in the Baltic Proper where less DIP in the water column led to less N fixation and organic matter production, which in turn led to less organic matter input to sediments, less heterotrophic oxygen consumption, less hypoxia and thereby further increased P sequestration in the
oxygenated sediments. Also in the Gulf of Riga, where hypoxia was rare, the bioturbation–induced reduction of pelagic DIP had large effects on primary production and especially N–fixation. In the two runs with bioturbation there was no or very little pelagic DIP surplus available for the N–fixing phytoplankton group in contrast to the run without bioturbation, where N fixation added on average 0.97 g N m$^{-2}$ year$^{-1}$ or ca 17 000 tonnes N year$^{-1}$ to the basin. It should be noted that these bioturbation effects are only valid where conditions are favourable for P binding to metal oxides, which occur primarily in freshwater and
brackish sediments with a high iron and low sulphide content (Van Helmond et al., 2020; Jordan et al., 2008).

The significant effect of bioturbation on P retention found here is in line with the results of studies on the effects of the invasive polychaete *Marenzelleria* spp. in the Gulf of Finland and Stockholm archipelago in the northern Baltic Proper (Isaev et al., 2017; Norkko et al., 2012). However, both of these studies focussed on areas with very high abundances of *Marenzelleria* spp., and there are some indications that lower abundances may yield an opposite effect, i.e. increase P outflux from the sediment
(Norkko et al., 2012; Nyström Sandman et al., 2018). While these studies concentrated on a single taxon in a limited area, we included a dynamic representation of the whole benthic fauna community over the whole Baltic Sea. Further, these studies only included the effects of bioturbation, but excluded metabolic fluxes. Here, we estimated that even though the excretion of DIP by benthic fauna constituted a significant proportion of benthic–pelagic fluxes, about ¼ to ⅓, it did not reverse the effect of bioturbation on P retention. These results are supported by Berezina et al. (2019), who found a positive correlation between
macrofaunal biomass and P excretion, but a negative correlation between biomass and total sediment to water DIP flux in sediment cores with natural macrofauna communities in the eastern Gulf of Finland.

The effects of bioturbation on sediment N dynamics were less important for eutrophication processes than the effects on P dynamics in this study. We assumed a very simple process formulation, where bioturbation increases oxygen penetration depth in the sediments leading to a larger proportion of organic N mineralization in oxic environments, thus promoting outflux of
nitrates over benthic denitrification. In reality, denitrification is a complex process depending on e.g. the 3D–structure of redoxclines in the sediment. If the biogenic structures of tube–dwelling bio–irrigators increase the area of oxic–anoxic interface in the sediment, this can lead to the opposite effect where a larger proportion of nitrate is denitrified at the enlarged redoxcline (Aller, 1988; Gilbert et al., 2003). However, we believe this to be a special case unlikely to dominate in the Baltic Sea. Henriksen et al. (1983) measured an increased proportion of nitrate denitrified in sediments with large burrows of animals with





low irrigation activity (e.g. *Arenicola marina*), but a decrease in sediments with species common in the Baltic Sea (e.g. *Limecola balthica* and *Mya arenaria*).

To better capture alterations in redoxclines, a depth–resolved sediment model with oxygen as a state variable would be needed. We also recognize that many other possible effects of bioturbation, e.g. on burial (Josefson et al., 2002) and resuspension (Cozzoli et al., 2021) were not included. However, there is always a trade-off between model complexity and generality, with

few models to date combining a depth-resolved sediment module together with a full pelagic model (Ehrnsten et al., 2020b; Lessin et al., 2018). One of the main advantages of the BALTSEM model is that its simplicity and fast running time promotes the development of additional features and experimentation with a large number of simulations (e.g. Gustafsson et al., 2015; Soerensen et al., 2016; Undeman et al., 2015).

Experimental studies from the Baltic Sea report a range of positive, negative or negligible effects of benthic fauna on benthic-

pelagic DIN fluxes and denitrification rates, showing that these processes are highly context-dependent (Griffiths et al., 2017 and references therein). Studies with *Limecola balthica*, the dominating group in our simulations, likewise report increasing, decreasing and inverted nitrate fluxes compared to bare sediments (Stief, 2013 and references therein). On the other hand, ammonium outfluxes were consistently increased by macrofauna in 31 studies reviewed by Stief (2013), supporting our simulation results that animal metabolism plays a significant role in benthic-pelagic DIN fluxes, with ammonium excretion

constituting about half of the total DIN fluxes in 2000-2020.

### 4.3 Benthic-pelagic coupling in a changing environment

The Gulf of Riga had higher simulated benthic stocks and fluxes than the Baltic Proper. This can partly be attributed to the slightly higher primary production (171±19 vs 159±25 g C m$^{-2}$ year$^{-1}$ in 2000-2020), but probably more importantly to the shallower mean depth causing a larger proportion of pelagic production to sink to the sediments before it is mineralized in the

water column. In the Gulf of Riga, on average one third (32 %) of primary production reached the bottom as POC during 2000–2020, compared to one fifth (21 %) in the Baltic Proper.

Besides depth, the amount of organic matter export also depends on e.g. the type of plankton and temperature (Tamelander et al., 2017). Despite large interannual variations, there was a clear decreasing trend in the proportion of primary production exported to the seafloor of about 5 percentage points per decade in the Baltic Proper during 1970–2020 ($R^2 = 0.68$, F=103, p

< 0.0001). This coincides with an increase in water temperature and shift in phytoplankton composition towards an increased proportion of cyanobacteria, seen both in these simulations and in reality (Belkin, 2009; Kahru and Elmgren, 2014). A less clear decreasing trend of 2 percentage points per decade was simulated in the Gulf of Riga ($R^2$=0.29, F=20, p < 0.0001).

The proportion of primary production arriving at the seafloor also varied with nutrient loads, constituting almost half of the annual primary production in the reduced load BSAP scenario (47–48 %) compared to 21–27% in the HIGH load scenario.

Simultaneously, the amount of benthic fauna decreased and mineralization of sediment organic matter became more dominated

by microbial processes with reduced loads. Thus, we can conclude that both the absolute amount and the relative proportion of POM input mineralized by macrofauna increased with nutrient loads, but in relation to primary production the proportion mineralized by fauna was almost independent of changes in loads because of the opposite responses of sinking organic matter and fauna.

**5. Conclusions and outlook**

Using a newly developed modelling tool, significant effects of benthic macrofauna on C, N and P cycling were simulated in the semi-enclosed brackish-water Baltic Sea, with impacts on the ecosystem from the extent of hypoxic bottoms to the rates of pelagic nitrogen fixation and primary production. The magnitude of effects was dependent on depth and productivity, as shown by the comparison of two basins and different nutrient load scenarios. Even though these large-scale simulations contain
a large degree of uncertainty, they are an important complement to empirical studies, which for practical reasons can only consider temporally and spatially limited parts of the system (Boyd et al., 2018; Snelgrove et al., 2014). Our results suggest that in addition to the much-studied bioturbation, the metabolism of benthic fauna should be given more attention in future studies as it may play a significant role in benthic mineralization of organic C, N and P in coastal seas and estuaries.

These simulations confirm the notion that benthic–pelagic coupling is strongest in shallow coastal areas (Griffiths et al., 2017;
Nixon, 1981), but also show that this relationship is modified by multiple physical and biological drivers, which may change over time. Unravelling these interacting drivers and responses on a system scale is important to understand how coastal and global biogeochemical cycles are responding to changes in, e.g. nutrient loads and climate.

**Code availability.** The model code is available upon request from the corresponding author.

**Author contributions.** All authors contributed to experimental design and interpretation of results. EE and BG developed the model code. EE performed the experiments, analysed results and wrote the manuscript with significant contributions from BG and OS.

**Competing interests.** The authors declare that they have no conflict of interest.

**Acknowledgements.** This study was supported by The Swedish Agency for Marine and Water Management through their
grant 1:11 – Measures for marine and water environment. We thank Bärbel Müller-Karulis for the development of and help with implementation of a script for relative bias calculations and Alexander Sokolov for access to environmental validation data through the BED database. Erik Smedberg is acknowledged for contributions to artwork (Figure 1).




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





**Figures and tables**

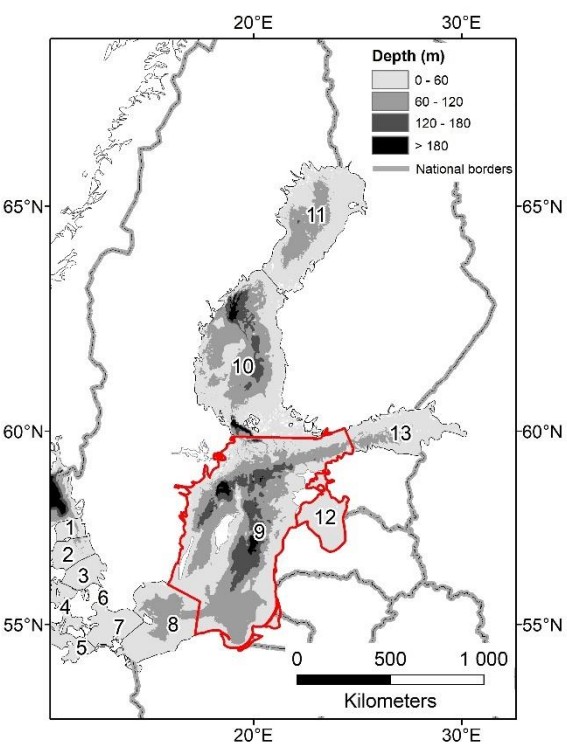

**Figure 1.** The Baltic Sea hypsography with a depth step of 60 m and basin divisions in the BALTSEM model. This study focusses on the
Baltic Proper (basin 9) and Gulf of Riga (basin 12), outlined in red.



**Figure 2.** Schematic overview of benthic model processes shared by benthic C, N and P components (a) and apportionment of mineralization fluxes of sediment N (b) and P (c) with bioturbation effects indicated in red.



**Figure 3.** Comparison of simulated total biomasses of benthic fauna to observations at four depth intervals in six basins from south to north. Observations are shown both as the sum of the three functional groups *L. balthica*, surface deposit-feeders and predator/scavengers ('Data L+D+P') and total observed fauna. All data are shown as means ± standard deviations of 1990-2012, except for Arkona Basin 0-30 m where observational data from 1965-1979 were used as no other data were available. Numbers after basin names refer to basin numbers in Fig. 1. Numbers of samples and further comparisons are presented in Appendix C.



**Figure 4.** Average (2000-2020) benthic fluxes (g m$^{-2}$ day$^{-1}$) and stocks (g m$^{-2}$) of C (a, b), N (c, d) and P (e, f) in the Baltic Proper (0–90 m, left column) and Gulf of Riga (right column). Arrow widths are proportional to fluxes for each element. Seq. = P sequestration.



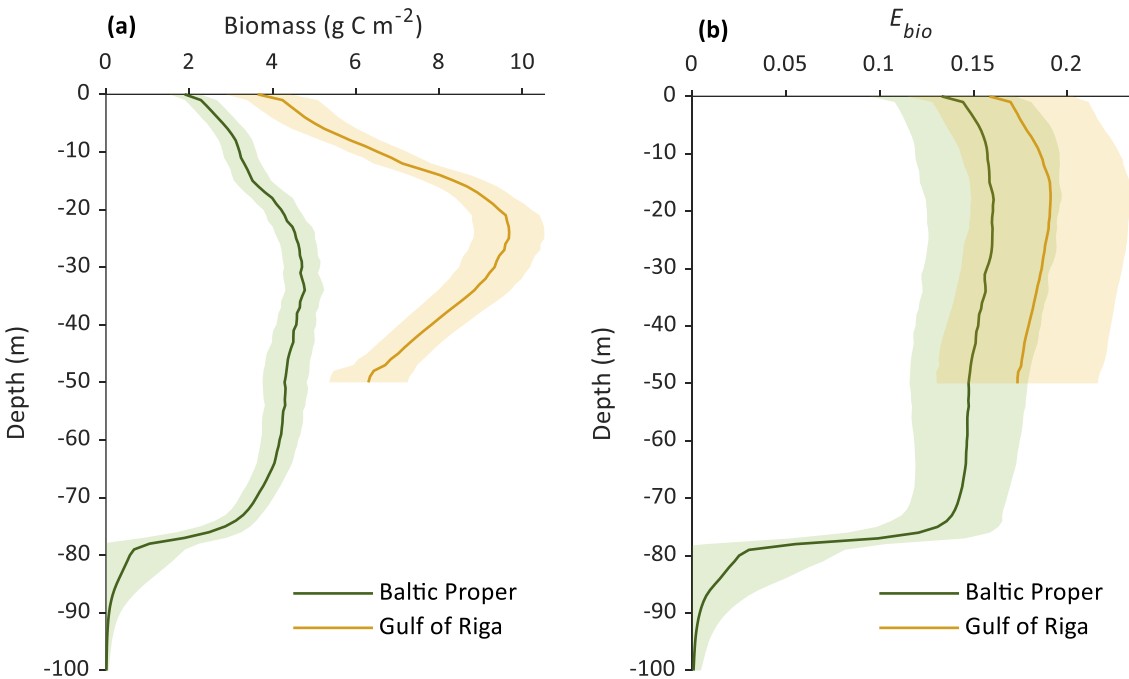

**Figure 5.** Depth distribution of benthic fauna biomass and the bioturbation coefficient $E_{bio}$ in the upper 100 m of the Baltic Proper and Gulf of Riga. Averages (lines) and standard deviations (shaded areas) of biweekly values 2000-2020 in the default model run ($E_{max} = 0.3$).

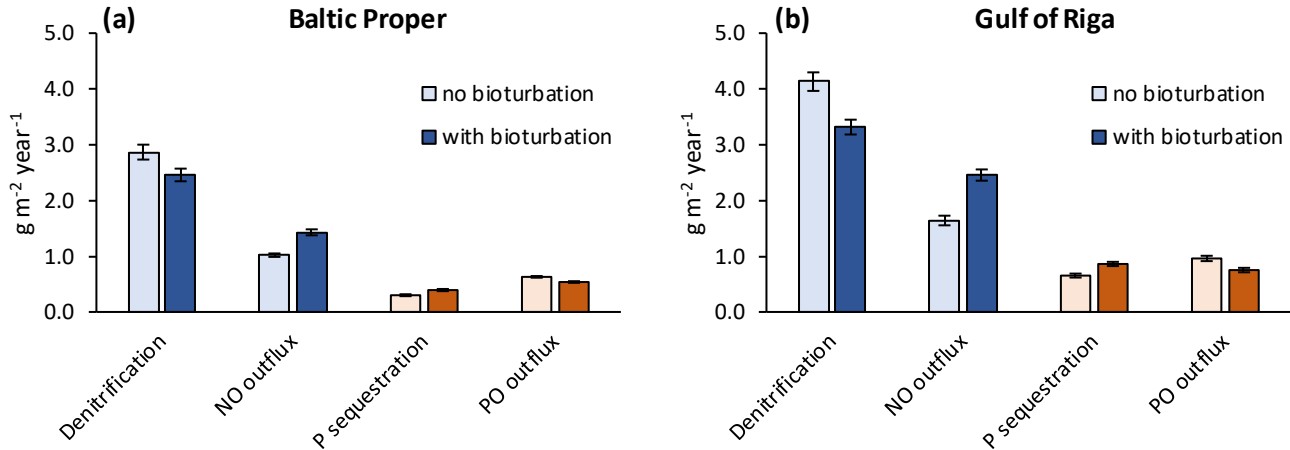

**Figure 6.** Direct effects of bioturbation on benthic fluxes. Benthic fluxes directly affected by bioturbation in the default run with bioturbation and when calculated for each time-step without bioturbation. Averages for 2000–2020 ± standard deviations in the Baltic Proper (0–90 m depth, a) and Gulf of Riga (b). Note that 'outflux' refers to the flux from sediments to the water column without animal excretion.





**Figure 7.** Sensitivity analysis of direct and indirect effects of three levels of bioturbation on benthic fluxes of carbon (a, b), nitrogen (c, d) and phosphorus (c, d). Averages for 2000–2020 ± standard deviations in the Baltic Proper (0–90 m depth, left column) and Gulf of Riga (right column). Note that animal excretion is shown separately and not included in 'mineralization' or 'outflux'.




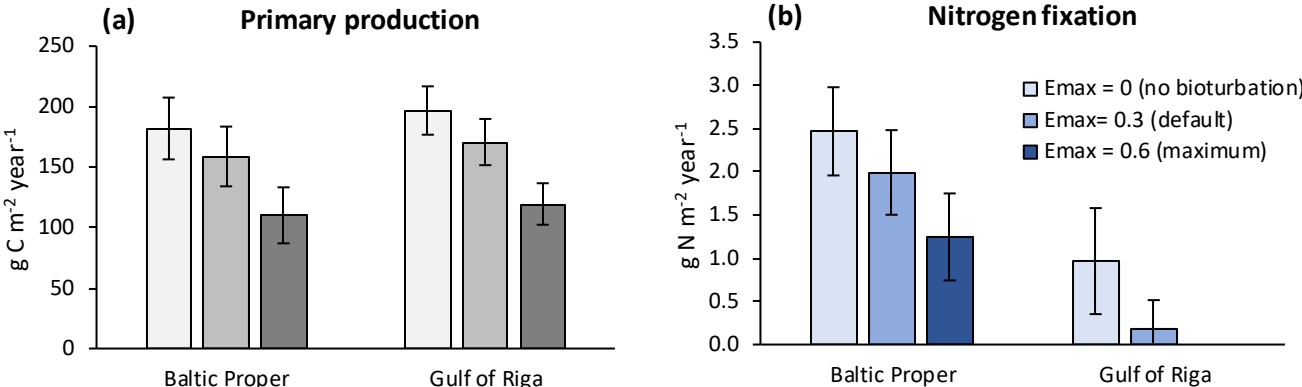

Figure 8. Sensitivity analysis of effects of three levels of bioturbation on total primary production (a) and N fixation by cyanobacteria (b). Averages for 2000–2020 ± standard deviations in the Baltic Proper (0–90 m depth) and Gulf of Riga.

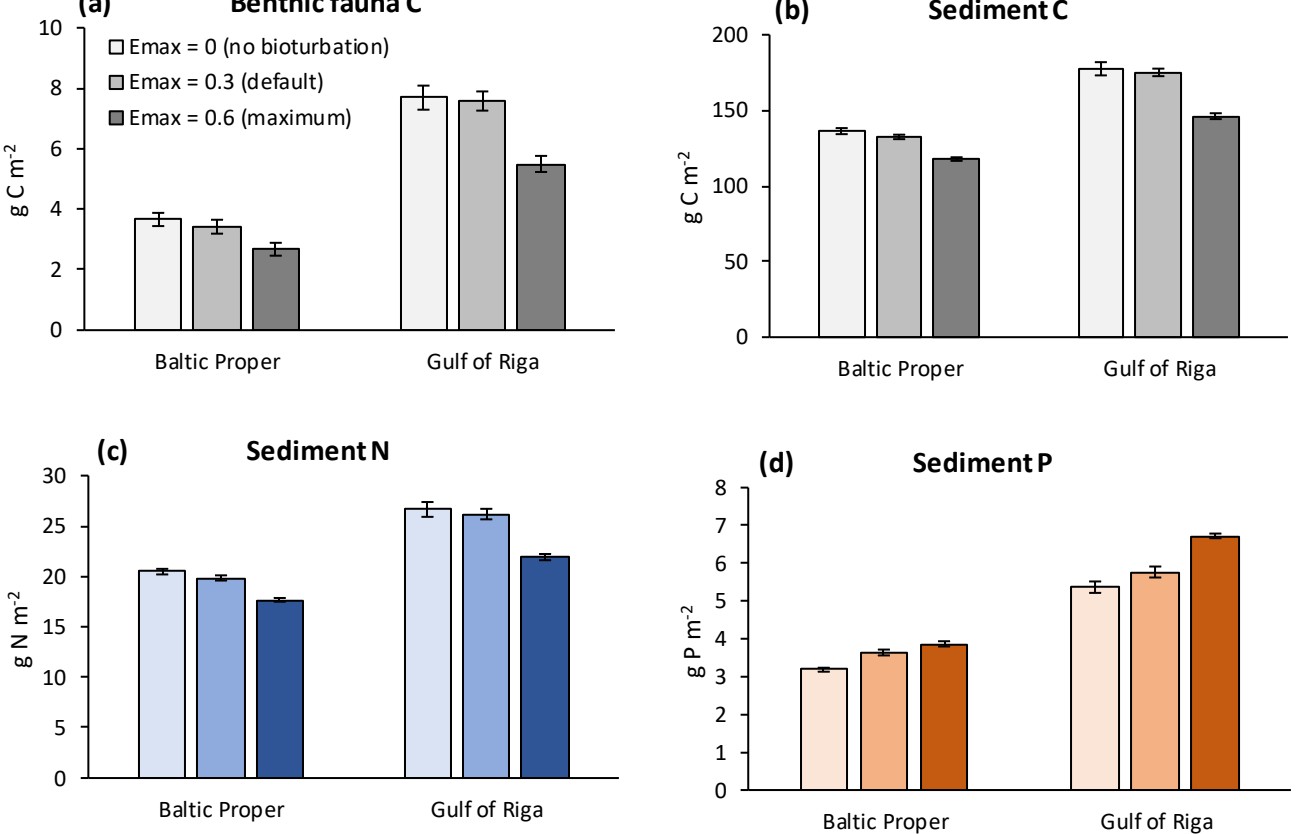

Figure 9. Sensitivity analysis of effects of three levels of bioturbation on stocks of benthic fauna and sediment C, N and P. Averages for 2000–2020 ± standard deviations in the Baltic Proper (0–90 m depth) and Gulf of Riga.





**Figure 10.** Input of POC (a), PON (b) and POP (c) to the sediment, stocks of benthic fauna (d), bioturbation enhancement coefficient (e) and hypoxic area (f) in the default model run 2000-2020 and in two nutrient load scenarios 2080-2100 in the Baltic Proper (0–90 m depth) and Gulf of Riga. The hypoxic area is defined as the annual maximum extent of areas with oxygen concentration < 2 mg $O_2$ $l^{-1}$ and is given for the whole basins on a logarithmic scale (f).



**Figure 11.** Apportionment of benthic fluxes of carbon (a, b), nitrogen (c, d) and phosphorus (c, d) in the default model run 2000-2020 and in two nutrient load scenarios 2080-2100 in the Baltic Proper (0–90 m depth, left column) and Gulf of Riga (right column). Fluxes are shown as percent of POM input to the sediment. Note that animal excretion is shown separately and not included in 'mineralization' or 'outflux'.





**Table 1.** Comparison of simulated benthic fauna biomass in the Gulf of Riga and estimates based on field sampling (g wet weight m$^{-2}$).

| Total (g wwt m$^{-2}$) | Range (g wwt m$^{-2}$) | Source | Comment |
|---|---|---|---|
| 154 | 29 to 284 | Model | 1970–2020 |
| 64 | <2 to >300 | (Gogina et al., 2016) | |
| 46 | 38 to 200 | (Järvekülg, 1983) | Unit uncertain, given as g m$^{-2}$ |
| 350 | 160 to 370 | (Kotta et al., 2008) | Assuming 10% dwt wwt$^{-1}$ |
| 38 | 1 to 188 | (Witek, 1995) | SW part only |



**Appendix A. Mathematical description of benthic model dynamics in BALTSEM-BMM**

All state variables are listed in Table A1 and benthic parameters in Tables A2-A4. A graphical overview of benthic state variables and processes is provided in the main manuscript (Fig. 2). Bioavailable surface sediments are represented as terraces at 1 m depth intervals with and area corresponding to the hypsography of each basin. All benthic state variables are calculated in mg m$^{-2}$ for each terrace.

**Sediment dynamics**

Sediment concentrations of elements X (X = C, N, P) are divided into three banks that share the same processes (Eq. A1–A3). Deposition of sinking detritus ($D_{DETX}$) and phytoplankton ($D_{PHYiX}$, $i$ = phytoplankton group 1, 2, 3) on the sediment is integrated into a bank of fresh organic matter SED1X (Eq. A1). Loss terms of the bank are mineralization ($Z_{SED1X}$), aging ($A_{SED1X}$) into the second food bank SED2X and uptake by deposit-feeders ($U_{SED1X,BF2}$). Sources of SED2X include aging from bank 1 ($A_{SED1X}$) and faeces from deposit-feeders ($F_{BF2X}$) and predators ($F_{BF3X}$) (Eq. A2). Loss terms are mineralization ($Z_{SED2X}$), uptake by

*Limecola balthica* ($U_{SED2X,BF1}$) and aging ($A_{SED2X}$) into the third sediment bank SED3X, which is considered unavailable as food for the benthic fauna, but available for bacterial mineralization. Mortality ($M_{BFiX}$) of all benthic groups and faeces of *L. balthica* are added to the third bank (Eq. A3). Loss terms are mineralization ($Z_{SED2X}$) and burial ($B_x$).

$$\frac{dSED1X}{dt} = \sum_{i=1}^{3}(D_{PHYiX}) + D_{DETX} - Z_{SED1X} - A_{SED1X} - U_{SED1X,BF2} \tag{A1}$$

$$\frac{dSED2X}{dt} = A_{SED1X} + F_{BF2X} + F_{BF3X} - Z_{SED2X} - A_{SED2X} - U_{SED2X,BF1} \tag{A2}$$

$$\frac{dSED3X}{dt} = A_{SED2X} + F_{BF1X} + \sum_{i=1}^{3}(M_{BFiX}) - Z_{SED3X} - B_X \tag{A3}$$

Sediment silica is modelled as a single pool with sinking diatom Si (PHY2Si) as a source and mineralization and burial as sinks (Eq. A4).

$$\frac{dSEDSi}{dt} = D_{PHY2Si} - Z_{SEDSi} - B_{Si} \tag{A4}$$

Mineralization (Eq. A5) and aging (Eq. A6) of element X (X = C, N, P) in bank SED$i$ ($i$ = 1, 2, 3) are formulated as first-order

reactions, with the rate constants $a_{ZSEDX}$ and $k_{SEDi}$, respectively. Mineralization and aging are temperature-dependent according to the functions $Q_{T1}$ (Eq. A38) and $Q_{T2}$ (Eq. A39), respectively.

$$Z_{SEDiX} = a_{ZSEDX} Q_{T1} SEDiX \tag{A5}$$

$$A_{SEDiX} = k_{SEDi} Q_{T2} SEDiX \tag{A6}$$

A proportion $a_{BSEDX}$ of the third sediment bank is buried (Eq. A7).

$$B_X = a_{BSEDX} SED3X \tag{A7}$$





For each element X (X = C, N, P), the mineralization fluxes from all three sediment banks are summed into a total flux ($Z_{SEDXtot}$), which is further divided in the same way as in the standard BALTSEM with the addition of bioturbation effects. For C, the total sediment mineralization flux goes to the pelagic dissolved inorganic carbon (DIC) pool ($O_{DIC}$, Eq. A8).

$$O_{DIC} = Z_{SEDCtot} \tag{A8}$$

Depending on oxygen concentrations in the bottomwater layer (OXY), mineralized sediment N is released to the water column as ammonia ($O_{NH}$, Eq. A9), oxidised N ($O_{NO}$, Eq. A10) or denitrified to $N_2$ ($W_{Deni}$, Eq. A11). NH represents total ammonia ($NH_3$ and $NH_4$) and NO the sum of nitrite ($NO_3$) and nitrate ($NO_4$).

$$O_{NH} = v_{NOXY} Z_{SEDNtot} \tag{A9}$$

$$O_{NO} = (1 - v_{NOXY})\eta_N Z_{SEDNtot} \tag{A10}$$

$$W_{Deni} = (1 - v_{NOXY})(1 - \eta_N)Z_{SEDNtot} \tag{A11}$$

In anoxic or nearly anoxic conditions, the mineralized N is released to the water column as NH, defined by the fraction $v_{NOXY}$ (Eq. A12), otherwise it is oxidized.

$$v_{NOXY} = \left(1 + \frac{(OXY + |OXY| + a_{vNOXY})}{(|OXY| + a_{vNOXY})^8}\right)^{-1} \tag{A12}$$

Subsequently, the oxidized fraction can be denitrified into $N_2$ (treated as a permanent sink) or released to the pelagic NO pool.
The fraction released as NO ($\eta_N$,) is positively related to oxygen concentrations according to Eq. (A13), where $q_N$, $a_N$, $b_N$ and $c_N$ are fitting constants (Savchuk, 2002) and $E_{bio}$ is a bioturbation enhancement factor (see Eq. (A37) below).

$$\eta_N = q_N \left(\frac{1}{1+exp\left(a_N-(b_N + E_{bio})\left(c_N+max(OXY,0)\right)\right)} - \frac{1}{1+exp(a_N-b_N c_N)}\right) \tag{A13}$$

A fraction $\eta_P$ of mineralized sediment P is sequestered ($K_P$, Eq. A14), i.e. bound to iron oxides in the sediment, while the rest is released to the pelagic phosphate (PO) pool ($O_{PO}$, Eq. A15).

$$K_P = \eta_P Z_{SEDPtot} \tag{A14}$$

$$O_{PO} = (1 - \eta_P)Z_{SEDPtot} \tag{A15}$$

The fraction sequestered is positively related to oxygen concentrations (enhanced by bioturbation) and negatively related to salinity according to Eq. (A16), where $q_P$, $a_P$, $b_P$, $c_P$, $d_P$, $e_P$, $f_P$ and $g_P$ are fitting constants. The fraction has an upper limit of 1 (100% of mineralized P sequestered), but can take on negative values, representing a release of previously sequestered P in
severely hypoxic or anoxic conditions. The salinity (SAL) dependence in the third term is used as a proxy for the higher availability of iron in the low-saline Bothnian Bay.





$$\eta_P = q_P \tanh(a_P OXY) + \frac{b_P (1 + E_{bio}) \max(OXY, 0)}{c_P + \max(OXY, 0)} - d_P \left(1 + e_P \exp\left(\frac{f_P - SAL}{g_P}\right)\right)^{-1} \tag{A16}$$

Sediment C mineralization and nitrification consume oxygen, while denitrification causes reimbursement of $O_2$. Thus, the sediment consumption of bottom water oxygen, $O_{SEDOXY}$, is calculated according to Eq. (A17). The constants $\beta_{ZC}$, $\beta_{Nit}$ and $\beta_{Deni}$
are oxygen equivalents for the respective processes.

$$O_{SEDOXY} = \beta_{ZC} Z_{SEDCtot} + \beta_{Nit}(1 - v_{NOXY}) Z_{SEDNtot} - \beta_{Deni} W_{Deni} \tag{A17}$$

**Benthic fauna**

The model includes three functional groups of benthic fauna: *Limecola balthica* (BF1), deposit-feeders (BF2) and predators (BF3) that share the same processes but with group-dependent parameterisations. The change in C biomass of functional group
$i$ ($i = 1, 2, 3$) is the difference between food uptake, faeces production, respiration, mortality and predation (Eq. A18), where $U_{BFiC,BF3}$ denotes predation on group $i$ ($i = 1, 2$) by benthic predators.

$$\frac{dBFiC}{dt} = U_{BFiC} - F_{BFiC} - R_{BFiC} - M_{BFiC} - U_{BFiC,BF3} \tag{A18}$$

Since the model does not include recruitment or migration, biomass change is set to 0 when biomass falls below 0.01 mg C $m^{-2}$ to avoid permanent extinction of any group.

The change in N and P components of a group BF$i$X ($i = 1, 2, 3$; X = N, P) share the same processes as above, except that respiratory release of C is replaced by excretion of N or P (Eq. A19). As the fauna has fixed stoichiometry, the dynamics can also be expressed by the change in C biomass and a conversion factor $\lambda_{CX}$.

$$\frac{dBFiX}{dt} = U_{BFiX} - F_{BFiX} - E_{BFiX} - M_{BFiX} - U_{BFiX,BF3X} = \frac{dBFiC}{dt} \lambda_{CX}^{-1} \tag{A19}$$

Food uptake of element X (X = C, N, P) by group $i$ is the sum of ingestion of food sources $j$, where $I_{jX,BFi}$ is ingestion rate of
food source $j$ by group $i$ (Eq. A20).

$$U_{BFiX} = \sum_{j=1}^{n} (I_{jX,BFi} BFiX) \tag{A20}$$

Predators feed on deposit-feeders and *L. balthica,* with a strong preference for the former. The ingestion rate of multiple food sources is formulated according to Eq. (A21), where $I_{jC,BF3}$ is the ingestion rate of food source $j$ by predators, $I_{max,BF3}$ is the maximum specific ingestion rate of predators, $pr_j$ and $pr_k$ are preference factors for food sources $j$ and $k$, $Flim_j$ and $Flim_k$ and
lower feeding limits on $j$ and $k$ and $K_{m,BF3}$ is a half-saturation constant for predator ingestion rate.

$$I_{jC,BF3} = I_{max,BF3} \left(\frac{pr_j (jC - Flim_j)}{K_{m,BF3} + \sum_{k=1}^{n} (pr_k (jC_k - Flim_k))}\right) Q_{T1} \tag{A21}$$





Deposit-feeders are restricted to feeding on freshly deposited organic matter in SED1X, while *L. balthica* can eat slightly older organic matter in SED2X. *L. balthica* switches to suspension-feeding when phytoplankton concentrations are high (>2 mg Chl-a m$^{-3}$). Ingestion rate of a single food source $j$ by group $i$ can be simplified to Eq. (22):

$$I_{jC,BFi} = I_{max,BFi} \left( \frac{jC}{K_{m,BFi} + jC} \right) Q_{T1} \tag{A22}$$

Additionally, feeding stops at anoxia for all groups.

The ingestion rate of food source component X (X = N, P) is proportional to the C:X ratio of the food source $j$ (Eq. A23).

$$I_{jX,BFi} = I_{jC,BFi} \frac{jX}{jC} \tag{A23}$$

Ingested food is divided into assimilated uptake ($AU_{BFiX}$, Eq. A24) and feces ($F_{BFiX}$, Eq. A25) by an assimilation factor $AF_j$, which depends on the assumed nutritional quality of the food source $j$.

$$AU_{BFiX} = \sum_{j=1}^{n} (AF_j \, I_{jX,BFi} BFiX) \tag{A24}$$

$$F_{BFiX} = \sum_{j=1}^{n} \left( (1 - AF_j) \, I_{jX,BFi} BFiX \right) \tag{A25}$$

Respiratory release of DIC ($R_{BFiC}$) is divided into three terms: basal respiration related to biomass and temperature, growth and activity respiration related to food uptake, and possible excess respiration to keep stoichiometry ($Rex_{BFiC}$) according to Eq. (A26), where $r_{b,BFi}$ and $r_{g,BFi}$ are the basal and growth respiration constants of group $i$, respectively.

$$R_{BFiC} = r_{b,BFi} Q_{T1} BFiC + U_{BFiC} r_{g,BFi} + Rex_{BFiC} \tag{A26}$$

Excretion of N and P ($E_{Xi}$) is formulated in the same way as respiration, and adds NH or PO to bottom water, respectively (Eq. A28).

$$E_{BFiX} = r_{b,BFi} Q_{T1} BFiX + U_{BFiX} r_{g,BFi} + Eex_{BFiX} \tag{A28}$$

To calculate excess respiration or excretion, first the limiting element for growth ($lim_i$) is calculated by comparing the C:N:P stoichiometry of assimilated food uptake to the stoichiometry of the fauna group $i$ (Eq. A29), where $minloc$ refers to the location of the minimum term within the brackets.

$$lim_i = \begin{cases} C, & minloc(AU_{BFiC}, AU_{BFiN} \lambda_{CN}, AU_{BFiP} \lambda_{CP}) = 1 \\ N, & minloc(AU_{BFiC}, AU_{BFiN} \lambda_{CN}, AU_{BFiP} \lambda_{CP}) = 2 \\ P, & minloc(AU_{BFiC}, AU_{BFiN} \lambda_{CN}, AU_{BFiP} \lambda_{CP}) = 3 \end{cases} \tag{A29}$$

For the limiting element, excess respiration/excretion is 0. The other two elements are then released to restore stoichiometry of the fauna. Thus, total respiration and excretion of group $i$ is given by the matrix in Eq. (A30-A32).




$$IF\ lim_i = C \begin{cases} R_{BFiC} = r_{b,BFi}Q_{T1}BFiC + U_{BFiC}\,r_{g,BFi} \\ E_{BFiN} = \frac{R_{BFiC}}{\lambda_{CN}} + AU_{BFiN} - \frac{AU_{BFiC}}{\lambda_{CN}} \\ E_{BFiP} = \frac{E_{BFiP}}{\lambda_{CP}} + AU_{BFiP} - \frac{AU_{BFiC}}{\lambda_{CP}} \end{cases} \tag{A30}$$

$$IF\ lim_i = N \begin{cases} E_{BFiN} = r_{b,BFi}Q_{T1}BFiN + U_{BFiN}\,r_{g,BFi} \\ E_{BFiP} = \frac{E_{BFiP}\lambda_{CN}}{\lambda_{CP}} + AU_{BFiP} - \frac{AU_{BFiN}\lambda_{CN}}{\lambda_{CP}} \\ R_{Ci} = E_{BFiN}\lambda_{CN} + AU_{BFiC} - AU_{BFiN}\lambda_{CN} \end{cases} \tag{A31}$$

$$IF\ lim_i = P \begin{cases} E_{BFiP} = r_{b,BFi}Q_{T1}BFiP + U_{BFiP}\,r_{g,BFi} \\ E_{BFiN} = \frac{E_{BFiP}\lambda_{CP}}{\lambda_{CN}} + AU_{BFiN} - \frac{AU_{BFiP}\lambda_{CP}}{\lambda_{CN}} \\ R_{BFiC} = E_{BFiP}\lambda_{CP} + AU_{BFiC} - AU_{BFiP}\lambda_{CP} \end{cases} \tag{A32}$$

Respiration consumes bottom water oxygen according to Eq. (A33).

$R_{BFiO} = \beta_{RC}R_{BFiC}$ (A33)

Mortality is divided into hypoxia-induced mortality and other mortality. Other mortality rate $m_{other,i}$ is linear for *L. balthica* (Eq. A34) and quadratic for the other two groups (Eq. A35).

$M_{BF1X} = \left(m_{other,BF1} + m_{ox,BF1}\right)BF1X$ (A34)

$M_{BFiX} = m_{other,BFi}BFiX^2 + m_{ox,BFi}BFiX$ (A35)

The hypoxia-induced mortality rate $m_{ox,BFi}$ is dependent on bottom water oxygen concentration, temperature and the functional group's sensitivity to hypoxia according to Eq. (A36), where $m_{0,BFi}$ is the mortality rate at anoxia and 10°C and $K_{ox,BFi}$ is a hypoxic sensitivity constant (Timmermann et al., 2012).

$m_{ox,BFi} = \left(1 - m_{0,BFi}\right) \frac{m_{0,BFi}\,exp\left(-K_{ox,BFi}OXY\right)}{1 - m_{0,BFi}\,exp\left(-K_{ox,BFi}OXY\right)} Q_{T3}$ (A36)

**Bioturbation**

Similar to Isaev et al. (2017), we use simple formulations for the effects of bioturbation on sediment denitrification (Eq. A13) and P sequestration (Eq. A16) through a bioturbation enhancement factor $E_{bio}$ which mimics increased oxygen penetration into the sediment. $E_{bio}$ uses the feeding rate of fauna as a proxy of their bioturbation activity (Blackford, 1997) according to Eq. (A37), where $E_{max}$ is the maximum enhancement, $cf_{BFi}$ is a contribution factor of functional group $i$ ($i$ = 1, 2, 3), $U_{BFiC}$ is the carbon uptake rate of group $i$ and $K_{bio}$ is a half saturation constant. As *L. balthica* is more sedentary than the other groups, a

contribution factor of 0.5 was assigned to it and a factor of 1 to the two other groups (Ebenhöh et al., 1995; Gogina et al., 2017 and refences therein).





$$E_{bio} = E_{max} \left( \frac{\sum_{i=1}^{3}(cf_{BFi}U_{BFiC})}{\sum_{i=1}^{3}(cf_{BFi}U_{BFiC}) + K_{bio}} \right) \tag{A37}$$

**Temperature dependencies**

Temperature (T) dependencies in the equations above are formulated according to Eq. (A38-A40).

$$Q_{T1} = exp(b_{ZSEDX}T^2) \tag{A38}$$

$$Q_{T2} = Q_{10}^{(T-10)/10} \tag{A39}$$

$$Q_{T3} = Q_{10ox}^{(T-10)/10} \tag{A40}$$

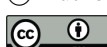



**Table A1.** State variables in BALTSEM-BMM.

| Short name | Long name | Unit |
| --- | --- | --- |
| *Benthic state variables* | | |
| SED1C | Sediment organic C bank 1 | mg C m$^{-2}$ |
| SED2C | Sediment organic C bank 2 | mg C m$^{-2}$ |
| SED3C | Sediment organic C bank 3 | mg C m$^{-2}$ |
| SED1N | Sediment organic N bank 1 | mg N m$^{-2}$ |
| SED2N | Sediment organic N bank 2 | mg N m$^{-2}$ |
| SED3N | Sediment organic N bank 3 | mg N m$^{-2}$ |
| SED1P | Sediment organic P bank 1 | mg P m$^{-2}$ |
| SED2P | Sediment organic P bank 2 | mg P m$^{-2}$ |
| SED3P | Sediment organic P bank 3 | mg P m$^{-2}$ |
| SEDSi | Sediment bioavailable Si | mg Si m$^{-2}$ |
| BF1 | Benthic fauna group 1, *Limecola balthica* | mg C m$^{-2}$ |
| BF2 | Benthic fauna group 2, Deposit-feeders | mg C m$^{-2}$ |
| BF3 | Benthic fauna group 3, Predators | mg C m$^{-2}$ |
| *Pelagic state variables* | | |
| SAL | Salinity | - |
| T | Temperature | °C |
| OXY | Dissolved oxygen | g O$_2$ m$^{-3}$ |
| NH | Total ammonia (NH$_4^+$ + NH$_3$) | mg N m$^{-3}$ |
| NO | Oxidized N (NO$_3^-$ + NO$_2^-$) | mg N m$^{-3}$ |
| PO | Total phosphate (H$_3$PO$_4$ + H$_2$PO$_4^-$ + HPO$_4^=$ + PO$_4^{3-}$) | mg P m$^{-3}$ |
| SiO | Dissolved Si (Si(OH)$_4$ + SiO(OH)$_3^-$) | mg Si m$^{-3}$ |
| DETN | Detrital N | mg N m$^{-3}$ |
| DETP | Detrital P | mg P m$^{-3}$ |
| DETSi | Detrital Si, biogenic Si | mg Si m$^{-3}$ |
| DETCm | Detrital C (autochthonous) | mg C m$^{-3}$ |
| DETCt | Detrital C (allochthonous) | mg C m$^{-3}$ |
| PHY1 | Phytoplankton group 1, N$_2$ fixers | mg N m$^{-3}$ |
| PHY2 | Phytoplankton group 2, diatoms | mg N m$^{-3}$ |
| PHY3 | Phytoplankton group 3, other phytoplankton | mg N m$^{-3}$ |
| ZOO | Zooplankton | mg N m$^{-3}$ |
| DONL | Labile dissolved organic N | mg N m$^{-3}$ |
| DONR | Refractory dissolved organic N | mg N m$^{-3}$ |
| DOPL | Labile dissolved organic P | mg P m$^{-3}$ |
| DOPR | Refractory dissolved organic P | mg P m$^{-3}$ |
| DOCLt | Labile dissolved organic C (allochthonous) | mg C m$^{-3}$ |
| DOCRt | Refractory dissolved organic C (allochthonous) | mg C m$^{-3}$ |
| DOCLm | Labile dissolved organic C (autochthonous) | mg C m$^{-3}$ |
| DOCRm | Refractory dissolved organic C (autochthonous) | mg C m$^{-3}$ |
| DIC | Dissolved inorganic C | μmol kg$^{-1}$ |
| ALK | Total alkalinity | μmol kg$^{-1}$ |





**Table A2.** Benthic parameters in the BALTSEM-BMM model taken from the standard BALTSEM (Gustafsson et al., 2014). Recalibrated parameter values are indicated in bold. Subscript indices C, N, P and Si refer to the respective elements. Where different elements share the same parameter value, the indices are listed in a row.

| Parameter | | Unit | Value | Comment |
|---|---|---|---|---|
| $a_{ZSEDC,N}$ | Sediment C and N mineralization rate at 0°C | day$^{-1}$ | **0.0004** | Reduced from standard BALTSEM to compensate for explicit mineralization by benthic fauna |
| $a_{ZSEDP}$ | Sediment P mineralization rate at 0°C | day$^{-1}$ | **0.0005** | |
| $a_{ZSEDSi}$ | Sediment Si mineralization rate at 0°C | day$^{-1}$ | 0.00015 | |
| $b_{ZSEDC,N,P,Si}$ | Temperature constant for sediment mineralization | °C$^{-1}$ | 0.005 | |
| $a_{BSEDC,N,P,Si}$ | Burial rate | day$^{-1}$ | 0.00006 | |
| $a_{vNOXY}$ | Fitting constant in Eq. (A12) | | 0.001 | |
| $q_{\eta N}$ | Fitting constant in Eq. (A13) | | 0.5 | |
| $a_{\eta N}$ | Fitting constant in Eq. (A13) | | 5.0 | |
| $b_{\eta N}$ | Fitting constant in Eq. (A13) | m$^3$ g O$_2^{-1}$ | 0.5 | |
| $c_{\eta N}$ | Fitting constant in Eq. (A13) | g O$_2$ m$^{-3}$ | **0.004** | Calibrated to account for addition of $E_{bio}$ in $\eta_N$ curve |
| $q_{\eta P}$ | Fitting constant in Eq. (A16) | | 0.3 | |
| $a_{\eta P}$ | Fitting constant in Eq. (A16) | | 1.0 | |
| $b_{\eta P}$ | Fitting constant in Eq. (A16) | | 0.8 | |
| $c_{\eta P}$ | Fitting constant in Eq. (A16) | | 2.0 | |
| $d_{\eta P}$ | Fitting constant in Eq. (A16) | | 0.6 | |
| $e_{\eta P}$ | Fitting constant in Eq. (A16) | | 4.0 | |
| $f_{\eta P}$ | Fitting constant in Eq. (A16) | | 4.5 | |
| $g_{\eta P}$ | Fitting constant in Eq. (A16) | | 0.2 | |
| $\beta_{ZC}$ | Oxygen equivalents for sediment C mineralization | g O$_2$ mg C$^{-1}$ | 0.00297 | |
| $\beta_{Nit}$ | Oxygen equivalents for sediment nitrification | g O$_2$ mg N$^{-1}$ | 0.0046 | |
| $\beta_{Denit}$ | Oxygen equivalents for sediment denitrification | g O$_2$ mg N$^{-1}$ | 0.003 | |





**Table A3.** Benthic parameters in the BALTSEM-BMM model taken from the carbon-based BMM (Ehrnsten et al., 2020a). Parameters are applied equally to C, N and P components.

| Parameter | | Unit | BF1 *Limecola balthica* | BF2 Deposit-feeders | BF3 Predators | Food banks |
|---|---|---|---|---|---|---|
| $I_{max}$ | Maximum specific ingestion rate at 10°C | day$^{-1}$ | 0.06* 0.02** | 0.09 | 0.09 | |
| $AF$ | Assimilation factor | | 0.5* 0.8** | 0.7 | 0.7 | |
| $K_m$ | Ingestion half-saturation constant | mg C m$^{-2}$ mg C m$^{-3}$ | 8000* 300** | 2000 | 500 | |
| $pr$ | Preference factor of predator for prey | | 0.01 | 0.9 | | |
| $Flim$ | Lower biomass limit for predation | mg C m$^{-2}$ | 30 | 30 | | |
| $r_b$ | Basal respiration or excretion rate at 10°C | day$^{-1}$ | 0.005 | 0.012 | 0.012 | |
| $r_g$ | Growth and activity respiration or excretion factor | | 0.2 | 0.15 | 0.2 | |
| $m_0$ | Anoxic mortality rate at 10°C | day$^{-1}$ | 0.081 | 0.69 | 0.069 | |
| $K_{ox}$ | Sensitivity to hypoxia | (mg O$_2$ L$^{-1}$)$^{-1}$ | 2.5 | 1.5 | 2.5 | |
| $m_{other}$ | Other mortality rate | day$^{-1}$ (mg C m$^{-2}$ day)$^{-1}$ | $1\times10^{-3}$ | $1\times10^{-5}$ | $1\times10^{-5}$ | |
| $wwt{:}C$ | Wet weight to C ratio | mg wwt mg C$^{-1}$ | 20 | 10 | 11 | |
| $Q_{10ox}$ | $Q_{10}$-value for hypoxia-induced mortality | | 2.6 | 2.6 | 2.6 | |
| $Q_{10}$ | $Q_{10}$-value for other rates | | 2.0 | 2.0 | 2.0 | |
| $k_{SED1}$ | Aging rate of sediment bank 1 at 10°C | day$^{-1}$ | | | | 0.025 |
| $k_{SED2}$ | Aging rate of sediment bank 2 at 10°C | day$^{-1}$ | | | | 0.02 |

*Deposit-feeding **Suspension-feeding

**Table A4.** New parameters in the BALTSEM-BMM model.

| Parameter | | Unit | Value | Comment |
|---|---|---|---|---|
| $E_{max}$ | Maximum bioturbation enhancement | | 0.3 | See text |
| $cf_{BF1}$ | Contribution factor to bioturbation | | 0.5 | See text |
| $cf_{BF2}$ | Contribution factor to bioturbation | | 1 | See text |
| $cf_{BF3}$ | Contribution factor to bioturbation | | 1 | See text |
| $K_{bio}$ | Half-saturation constant for bioturbation | mg C m$^{-2}$ day$^{-1}$ | 30 | Based on mean uptake by fauna in Ehrnsten et al. (2020a) |
| $\lambda_{CN}$ | C:N ratio of benthic fauna | mg C mg N$^{-1}$ | 6 | See footnote* |
| $\lambda_{CP}$ | C:P ratio of benthic fauna | mg C mg P$^{-1}$ | 70 | See footnote* |
| $\beta_{RC}$ | Respiratory quotient for benthic fauna | g O$_2$ mg C$^{-1}$ | 0.00266 | 1 mol mol$^{-1}$ (Brey, 2001) |

*Based on stoichiometric ratios measured from Baltic Sea benthic fauna (Carman and Cederwall, 2001; Cederwall and Jermakovs, 1999;
Hedberg et al., 2020; Kahma et al., 2020; Kumblad and Bradshaw, 2008; Lehtonen, 1996; S. Mäkelin & A. Villnäs, unpubl.).





**Appendix B. Validation of pelagic variables**

As a measure of model performance, we calculate the relative bias of simulated and observed long-term monthly means for some of the main pelagic state variables as described in Savchuk et al. (2012), but with data extended to 2015. The index compares model-data difference with variability in the data, giving an estimate of how well the model captures variability in 855 nature on seasonal, annual and decadal scales.

**Methods**

Observations of salinity, temperature, and concentrations of oxygen, ammonium, nitrate, phosphate and silicate were collected from the Baltic Sea Environment Database (BED) and other major data sources around the Baltic Sea such as IOW (Germany), NERI (Denmark), SYKE-FMI (Finland), and SHARK (SMHI, Sweden) databases. The full list of the data contributors can be 860 found at http://nest.su.se/bed/ACKNOWLE.shtml.

Basin-wide monthly time-series were prepared from available long-term observations in the following way. All the measurements found in monthly intervals over 1970-2015 for all frequently sampled water layers within every BALTSEM basin, i.e. usually at 5 m intervals for the top 20 m of the water column and $10 - 25$ m intervals for the deeper parts of basins, were pooled together and averaged. Coastal measurements, defined as being sampled within 12 nautical miles from the shore, 865 were excluded for all basins except the three Danish Straits basins, where the 12 nm coastal strip covers almost the entire basins. Measurements from several deep and isolated trenches in the northern Baltic Proper were excluded as they often display their own dynamics, asynchronous to that in the domain of the larger basin.

To emphasize both long-term changes and seasonality of variables, time series of a model-data difference of pairwise monthly means were used. Because the seasonal cycle is also reflected in monthly standard deviations, especially in the upper part of 870 the water column these differences were scaled with month-specific standard deviation $SD_m$. $SD_m$ was calculated as the standard deviation of data collected in month $m$ during the period $1970 - 2015$ for each available sampling depth. To remove any remaining outliers, the estimated monthly standard deviations were replaced by a spline smooth fitted by a GAM model. To avoid shifts due to some seasons being over-represented in the field data, in every basin the relative bias $RB_i$ at each sampling depth was calculated as an average of the twelve months in the annual cycle:

$$RB_i = \frac{1}{12} \sum_{m=1}^{12} \left( \frac{1}{n_{im}} \sum_{j=1}^{n_{im}} \frac{|M_{imj} - D_{imj}|}{SD_{im}} \right) \tag{B1}$$

where $m = 1..12$ denotes the month in the seasonal cycle, $n_{im}$ is the number of monthly data averages $D_{imj}$ available at depth $i$ in month $m$ of year $j$, and $M_{imj}$ is the model averages computed at sampling depth $i$ in month $m$ of the same year $j$. Thus, being based on monthly values computed from available data over the entire simulated time interval, the relative bias simultaneously characterizes several time scales: seasonal, interannual, and decadal.





**Results and discussion**

The BALTSEM-BMM performs very similarly to the standard BALTSEM. The average relative bias of the analysed variables over all basins is 1.40, which can be compared to a relative bias of 1.41 for the standard BALTSEM model.

The model captures variations in physical parameters (salinity and temperature) and oxygen concentrations with a relative bias of mostly less than 2 (Figure B1). Simulated ammonium (NH) concentrations are lower than measured in the well oxygenated Bothnian Sea (basin 10) and upper parts of the Bothnian Bay (basin 11), as all ammonium is oxidised to nitrate (NO) in the model under these conditions. Together with an underestimation of NO utilization in intermediate depth layers by phytoplankton, this results in an overestimation of NO concentration in the upper part of several basins, including the Gulf of Riga (basin 12). Variations in PO concentrations are well captured in the Baltic Proper (basin 9), but overestimated in the Bothnian Sea (basin 10) and Bothnian Bay (basin 11). A detailed discussion of performance and sources of errors can be found in Savchuk et al. (2012) and a comparison of BALTSEM to other similar models in Eilola et al. (2011).

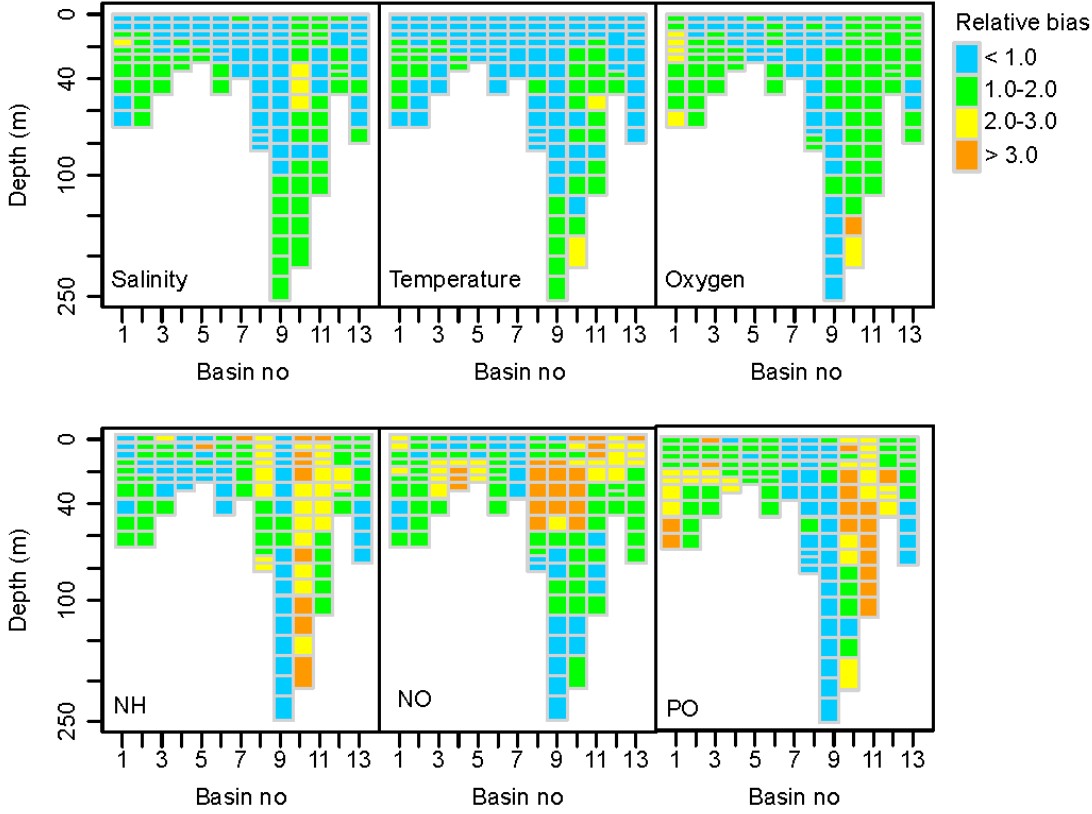

**Figure B1.** Spatial distribution of the relative bias between simulated and observed dynamics of salinity, temperature, and concentrations of oxygen, total ammonia (NH), nitrate + nitrite (NO) and total phosphate (PO) 1970-2015. See Fig. 1 for a map of basins.





**Appendix C. Validation of benthic fauna biomasses**

The simulated biomasses of benthic fauna were validated against observations using the method of Ehrnsten et al. (2020a), but with data extended to include the southern and southwestern parts of the Baltic Sea (basins 1-8 in Fig. 1). The comparison includes a visual comparison of biomasses over depth intervals in the different basins as well as a cost function assessment cf. Eilola et al. (2011).

**Methods**

As described in detail in Ehrnsten et al. (2020a), data from samples of benthic macrofauna biomass taken between 1990 and 2012 from the national databases of Sweden (www.sharkweb.smhi.se) and Finland (www.syke.fi/avointieto) as well as unpublished data gathered by the research vessel Aranda was used for validation. Only quantitative samples taken with a Van Veen or Smith McIntyre grab with an area of at least 0.1 m², sieved on a 1 mm mesh and weighed wet according to national standards were used. Additionally, samples containing hard substrates, missing biomass or depth data, or flagged as suspicious

in the database were removed from the dataset. Samples from areas defined as archipelago, embayment or river-dominated according to the EU Water Framework Directive were excluded, as the BALTSEM model does not represent these complex areas. The data was aggregated by basin and depth interval: 0–30 m, 30–70 m, 120–120 m and >120 m. As there was no data available for the Arkona Basin above 30 m in the time interval 1990–2012 or later, data collected by Aranda in 1965-1979 were used instead. The final validation dataset consisted of 7679 observations (Table C1).

The results of the default simulation were compared to observations using a cost function $CF=\left|(M-D)/SD\right|$ where M is model mean, D is mean of observations and SD is the standard deviation of observations (Eilola et al., 2011). According to Eilola et al. (2001), model results can be interpreted as good if the model mean is within one standard deviation of the observed mean ($0 \leq CF < 1$), reasonable if $1 \leq CF < 2$ and poor if $CF \geq 2$. Modelled carbon biomasses were converted to wet weight using the factors derived by Timmerman et al. (2012), presented in Table A3.

**Table C1.** Number of observations of benthic fauna wet weight used for model validation per basin and depth interval. Total N = 7679. Note that replicate samples are counted as individual observations.

| Depth interval | Kattegat (1-3) | Öresund (6) | Arkona Basin (7) | Bornholm Basin (8) | Baltic Proper (9) | Gulf of Finland (13) | Bothnian Sea (10) | Bothnian Bay (11) |
|---|---|---|---|---|---|---|---|---|
| 0-30 m | 864 | 623 | 6 | 412 | 292 | 10 | 408 | 144 |
| 30-70 m | 488 | 0 | 143 | 279 | 822 | 284 | 880 | 306 |
| 70-120 m | 65 | | | 49 | 330 | 78 | 479 | 486 |
| 120+ m | | | | | 58 | | 173 | |
| **Total** | **1417** | **623** | **149** | **740** | **1502** | **372** | **1940** | **936** |



## Results and discussion

The model is not applicable to the high-salinity areas at the entrance to the Baltic Sea (Kattegat and Öresund), as it does not
include the high diversity of functional groups present in these areas (Figs. C1–C2). In the Baltic Sea *sensu stricto* (basins 7–
13), the cost function (CF) values are mostly good to reasonable (Fig. C1), but a visual comparison shows that the standard
deviations of observations are very large in most cases, allowing for large deviations between modelled and observed means
(Fig. C2). However, the visual comparison also shows that the main observed trends in decreasing total biomass with increasing
latitude and depth are reasonably captured by the model, as well as the order of magnitude of the individual functional groups.

The closest match between simulated and observed means is seen in the Baltic Proper, which is the largest basin and which
contains the major part of benthic fauna stocks in the Baltic Sea. In the deepest segment of the Bornholm Basin (70-100 m),
observations are predominantly from hypoxic areas with biomasses of *L. balthica* close to 0, while simulations include some
oxic areas giving an average biomass of 0.6 g wwt m$^{-2}$, explaining the high CF values. In the Gulf of Finland, simulated
biomasses of *L. balthica* are considerably higher than observations. This might be due to the omission of possible negative
effects of low salinity on growth and/or reproduction, as discussed in Ehrnsten et al. (2020a). In the Bothnian Sea, conversely,
observed biomasses are higher than simulated ones for all groups, hinting at an underestimation of primary production and/or
sedimentation rates in this basin. In the oligotrophic Bothnian Bay, *L. balthica* is virtually extinct and total fauna biomasses
are an order of magnitude lower than in the other basins both in observations and in simulations.



| | Kattegat (1-3) | Öresund (6) | Arkona Basin (7) | Bornholm Basin (8) | Baltic Proper (9) | Gulf of Finland (13) | Bothnian Sea (10) | Bothnian Bay (11) | Cost function |
|---|---|---|---|---|---|---|---|---|---|
| **Total macrofauna** | | | | | | | | | 0.0 |
| 0-30 m | 5.33 | 4.05 | 1.88 | 1.40 | 0.48 | 0.92 | 0.72 | 0.50 | 0.5 |
| 30-70 m | 1.95 | ND | 0.75 | 0.43 | 0.16 | 1.52 | 0.78 | 0.31 | 1.0 |
| 70-120 m | 0.90 | | | 0.22 | 0.96 | 0.11 | 1.00 | 0.37 | 1.5 |
| 120+ m | | | | | 0.47 | | 0.84 | | 2.0 |
| *Limecola balthica* | | | | | | | | | 2.5 |
| 0-30 m | 78.63 | 8.91 | 1.81 | 1.49 | 0.46 | 1.15 | 0.71 | 0.18 | 3.0 |
| 30-70 m | NA | ND | 0.82 | 0.53 | 0.26 | 1.61 | 0.10 | 0.09 | |
| 70-120 m | NA | | | 23.24 | 1.22 | 1.91 | 0.07 | 0.04 | |
| 120+ m | | | | | 0.00 | | 0.15 | | |
| **Deposit-feeders** | | | | | | | | | |
| 0-30 m | 0.45 | 0.67 | 0.62 | 0.63 | 0.28 | 0.08 | 0.15 | 0.09 | |
| 30-70 m | 0.27 | ND | 0.03 | 0.29 | 0.55 | 0.35 | 0.83 | 0.11 | |
| 70-120 m | 0.65 | | | 0.29 | 0.28 | 0.57 | 0.92 | 0.44 | |
| 120+ m | | | | | 1.22 | | 0.76 | | |
| **Predators** | | | | | | | | | |
| 0-30 m | 0.27 | 0.35 | 1.07 | 0.25 | 0.27 | 0.84 | 0.34 | 0.65 | |
| 30-70 m | 0.51 | ND | 0.16 | 0.43 | 0.34 | 0.31 | 0.51 | 0.32 | |
| 70-120 m | 0.93 | | | 0.63 | 0.32 | 0.50 | 0.59 | 0.29 | |
| 120+ m | | | | | 0.68 | | 0.56 | | |


**Figure C1.** Cost functions comparing simulated and observed biomasses of benthic fauna. ND: no data. NA: Not applicable; in the deeper sections of Kattegat, observed biomasses of *L. balthica* are 0, i.e. CF cannot be computed.







**Figure C2**. Comparison of simulated biomasses of benthic fauna to observations at four depth intervals in BALTSEM basins from south to
north. Observations are shown both as the sum of the three functional groups *L. balthica*, surface deposit-feeders and predator/scavengers
('Data L+D+P') and total observed fauna ('Data all fauna'), including other groups such as suspension-feeders, freshwater herbivores and
large echinoderms. All data is given as means ± standard deviations of 1990-2012, except for Arkona Basin 0-30 m where data from 1965-
1979 was used as no other data was available. Numbers after basin names refer to basin numbers in Fig. 1. ND: no data.





**Figure C2. continued**







**Figure C2. continued**