# Peer review of "Large-scale effects of benthic fauna on carbon, nitrogen, and phosphorus dynamics in the Baltic Sea"

_Biogeosciences, 2022_

## Referee Comment (RC2)

Hans Cederwall
Baltic Bentos
Gethagsvägen 10
18463 Åkersberga
Sweden
Phone: +46 739832664
E-mail: hans.cederwall@telia.com

**Comments on the preprint "Large-scale effects of benthic fauna on carbon, nitrogen, and phosphorus dynamics in the Baltic Sea**

by Ehrnsten, Savchuk and Gustafsson.

This is a good paper trying to merge a benthic model with the older pelagic model by Savchuk and Wulff. The result is interesting, furthering modeling work is encouraged.

Remarks

Line 3, 22 and other places: Bioturbation in strictu means the mixing and turning over of the sediment. In the Baltic Proper that is dominantly carried out by the two amphipod species, who lies borrowed in the sediment during daytime, but swims about searching for food on the sediment surface during nighttime (they normally bioturbate the uppermost cm, down to ca 5 cm). To some part also Saduria takes place in the bioturbation but only on the suface layer. Species like Macoma, Halicryptus and Marenzelleria who are more or less permanently burrowed in the sediment (often deeper than 5 cm). They are more of bioirrigators. It should be mentioned early in the paper that the authours in their term bioturbation includes bioirrigation.

Line 65: Here is stated that the penetration depth of oxygen in Baltic Sea sediment is usually measured in mm rather than cm. This is only true for deeper bottoms from ca 60m or deeper. It could of course be found at shallower depth i archipelago areas, but those areas has been excluded in this work. I have measured redoxpotential at zoobenthos stations for more than 30 years and only on deep stations found oxygenation only in the top mm. The model here includes a depth interval of 0-120 m, so in the uppermost 60 m oxygenation is definitely better than what is said here.

Line 108: Limecola balthica has changed back to Macoma balthica (Caroline Raymond, pers. comm.)

Line 110: Pontoporeia is misspelled

Line 124: Misspellings: basalmaintenance  and  biomas s

Line 183: Why hindcast simulations for benthic variables.? There was a lot of data collected in the 1970s. For example in the Joint sampling programme (Elmgren 1978) and by revisting the Hessle-stations (Cederwall & Elmgren 1980). There should also be a lot of finnish data collected in the 1970s by Lassig and Andersin from the Finnish Institute of Marine Research. As far as I know these data were transferred to SYKE when the FIMR was closed down. Data from the Joint Sampling Programme and the Hessle project is stored in the benthic database BEDA. Contanct Caroline Raymond or Mats Blomqvist.

Line 213: I strongly suggest you leave at least the Arkona Basin out, beause the fauna here contains several other species than the ones you have mentioned in this preprint. Stick to the Gotland Basins, Gulf of Finland and Gulf of Riga who has a similar set of species.

Line 221, and Table 1: The benthic database BEDA contains primary data for 7 stations sampled in 1976 for the Joint Sampling Programme. In the mid 1990s a mapping of the macrofauna of the Gulf was done within the NMR financed Gulf of Riga Project. The results are published (Cederwall et al. 1999). Possibly the primay data are still held by Vadims Jermakovs, Latvian Institute of Aquatic Ecology (An institute where your colleague Bärbel Muller-Karulis earlier worked).

Line 277-279: Is the big difference in sedimentation between the BSAP scenario and the HIGH load scenario reflecting a difference in phytoplankton species composition? Historically the main input to benthic ecosystem has been the sedimentation of diatoms during the spring bloom, not the sedimentation during blue-green blooms. There has however been a shift in species composition in the spring bloom, where diatoms have decreased and other groups (who have lower sinking rate) have increased (Hjerne et al. 2019).

Line 397: cf Mäkelin & Villnäs 2022. Could the seasonal variations in benthic stoichiometry have any influence on your modelling work?

Fig 3: You show bars for the depth interval 0-30 m. I strongly suspect you have very few if any stations in the depth interval 0-10 m. This because you have outruled data from archipelagos and open sea research vessels don't like to go into shallow waters. Also these areas are dominated by transport bottoms hard to sample quantitately. Finally in these shallow bottoms you have another set of species than the set you have worked with. You should change to the depthinterval 10-30 m. In the text you mention that you had no shallower (<30 m) data from the Arkona Basin. The benthos database BEDA contains some data from the 1980s (Mats Blomqvist, pers. comm.) There is also a lot of data held by German institutes. But on the other hand I advise you to leave the Arkona Basin out of your paper since the fauna there differs so much from the Baltic Proper.

Table 1: Could becompleted with Cederwall et al. 1999.

Table A4, footnote: Mäkelin & Villnäs is published in Limnol. Oceanogr. 2022. The reference is missing in the reference list.

Line 901: You should know that sharkweb/SMHI does not have a benthological quality control of the data delivered to them. They assume that laboratories deliver correct data. This is not always the case (Mats Blomqvist pers. comm.). I suggest you import data from BEDA.

Final comment: The macrofauna is not the only part of the benthic ecosystem. The meiofauan for instance can have biomasses of 5-10 g wetweight/m2 (Elmgren 1976). To what extent does the meiofauna influence the sediment chemistry?

**Summing up**: This is basically a good paper well worth publishing after revision.

**References**

Cederwall & Elmgren 1980. Biomass increase of benthic macrofauna demonstrates eutrophication of the Baltic Sea. Ophelia, Suppl. I, 287-304.

Cederwall, Jermakovs & Lagzdins 1999.  Long-term changes in the soft-bottom macrofauna of the Gulf of Riga. ICES J. Mar. Sci. 55, Suppl. A, 41-48.

Elmgren 1976. Baltic benthos communities and the role of the meiofauna. Contrib. Askö Lab. 14, 31 pp.

Elmgren 1978.Structure and dynamics of Baltic benthos communities, with particular reference to the relationship between macro- and meiofauna. Kieler Meeresforsch. Sonderheft 4. 1-22.

Hjerne, Hajdu, Larsson, Downing & Winder 2019. Climate driven changes in timing, composition and magnitude of the Batic Sea phytoplankton spring bloom. Front. Mar. Sci. 6:482, 15 pp.

Mäkelin & Villnäs 2022. Food sources drive temporal variation in elemental soichiometry of benthic consumers. Limnol. Oceanogr. 9999, 1-16.

**Contact persons mentioned:**

Caroline Raymond
Dept. of ecology, environment and plant sciences
Stockholm University
10691 Stockholm
Sweden
Phone: +46 8164013
E-mail: caroline.raymond@su.se

Mats Blomqvist
HAFOK
Phone: +46 704135688
E-mail: mb@hafok.se

---

## Author Response (AR1)

**Response to reviewer comments RC1-3**

Please see our responses in blue below. Revised line numbers and figures are added in orange.

**Response to anonymous reviewer comments RC1**

**General comments**

The manuscript investigates the effects of benthic fauna on the biogeochemical transformations of major elements in the Baltic Sea via direct (metabolic) and indirect (bioturbation) pathways, by coupling a recently developed model of benthic fauna to an established hydrodynamic-biogeochemical model. Within regional ecosystem models, benthic components are often represented in a simplified way, where their role is largely limited to returning remineralised nutrients back into the water column. In the Baltic Sea, models accounting for indirect impacts of benthic fauna have been used for a while now, incentivised recently by the need to study the effects of bioturbation by invasive species. However, these models did not consider benthic faunal biomass explicitly, and thus did not allow a separation of effects of bioturbation and direct effects of faunal metabolism on benthic-pelagic dynamics. This manuscript bridges this gap and provides a detailed look at the role benthic fauna plays in shaping biogeochemical fluxes in the Baltic Sea. This study builds on the authors' previous work and contributes to the advancement of benthic system modelling, which has lagged behind its pelagic counterpart in coupled regional models. Although it is an important step in the right direction, I have several criticisms I would like to bring to the attention of the authors. Foremost they are related to the "large-scale" approach taken for this work. It should not serve as a substitute to thorough comparison of model results with data, nor to the detailed presentation and discussion of results (providing quantitative estimates and aligning text narrative with figures). Limitations of the model, unique contributions of this work to the understanding of the Baltic Sea biogeochemistry beyond state-of-the-art knowledge should be appropriately discussed. Below are my detailed comments and questions, that I hope will improve the quality of the manuscript.

We thank the reviewer for the comprehensive and detailed review of our manuscript. Based on the suggestions, we feel the manuscript has been significantly improved. We hope that our modifications of the manuscript detailed below, including additional validation and extended results and discussion, satisfies the reviewer.

**Specific comments**

**Title**

1. The title should clearly indicate that it is a model-based study.

We changed the title, removing the slightly ambiguous term "large-scale" and adding "modelling". The new proposed title is (lines 1-2):

Modelling the effects of benthic fauna on carbon, nitrogen, and phosphorus dynamics in the Baltic Sea

**Abstract**

1.   It would be informative to add quantitative estimates of changes due to direct and indirect contributions of benthic fauna. "Small proportion of seafloor organic stocks", "decreases denitrification", "increases P retention", "reduction in N fixation" etc all should be accompanied with quantitative estimates (% change). The way information is currently presented, it is hard to see what a unique contribution of this work is to the understanding of biogeochemical impacts of benthic fauna in the Baltic Sea, as the effects of bioturbation *via* sediment oxygenation on N, P dynamics in the Baltic Sea are generally well known from previous studies.

We added quantitative estimates to the abstract as suggested (lines 14-15).

2.   Line 16: It should be clear from the text that bioturbation affects denitrification, P retention etc *indirectly*, e.g., increasing oxygen penetration depth and availability.

We modified the sentence to clarify this (lines 16-18):

"Further, through enhanced sediment oxygenation, bioturbation decreases benthic denitrification and increases P retention, the latter having far–reaching consequences throughout the ecosystem."

3.   Line 20: "chain of indirect effects" is opposed to "direct effects of faunal respiration, excretion and bioturbation". This create some confusion over which effects are "direct" and which are "indirect" ones.

We realize the terms "direct" and "indirect" were used in a confusing way, and now reserve them for referring to metabolism as direct and bioturbation as indirect effects. Consequently, we modified the sentence to avoid confusion (line 20): "This chain of effects through the ecosystem overrides the local effects of faunal respiration, excretion and bioturbation." We made corresponding changes throughout the manuscript (lines 221-222, 272, 277 and Figures 6, 7).

**Introduction**

1.   Lines 43-44. "The combined effects of animal bioturbation and metabolism have seldom been studied together." Several references support this statement, but could the authors elaborate on what are the main reasons for that gap in research, despite recognition of the importance of benthic processes?

2.   Could you provide a more solid justification for your approach? Why do we need to implement modelling – does it address the knowledge gap? Why did biogeochemical models of the Baltic Sea so far did not include benthic fauna?

3.   When specifying reasons for using Baltic Sea as a model area, to what extent also relatively simple benthic community composition play role?

Response to comments 1.-3:

Several recent reviews and perspectives discuss the gap in research mentioned in the first comment, which essentially is due to the different research traditions, including

different foci, scales and assumptions in the several individual fields studying benthic processes (Ehrnsten et al., 2020b; Lessin et al., 2018; Middelburg, 2018; Snelgrove et al., 2014, 2018). We opened up some of the reasoning, and provide additional justification for why we need mechanistic modeling, why benthic fauna has not been included in previous models, and, as suggested in the third comment, why the Baltic Sea is and ideal system to develop this kind of model. Please see an extract of the introduction (lines 45-69) **with additions in bold** below:

"Even though the importance of benthic fauna for sediment biogeochemistry and benthic–pelagic fluxes has long been recognized (Rhoads, 1974), the combined effects of animal bioturbation and metabolism have seldom been studied together (Ehrnsten et al., 2020b; Middelburg, 2018; Snelgrove et al., 2018). **A long-standing assumption in biogeochemical sediment research is that animals contribute considerably to transport of solids and solutes through bioturbation, but their consumption of organic matter is of minor importance (Middelburg, 2018). However, several studies show that this assumption does not hold in many shallow coastal systems, as recently reviewed by Middelburg (2018) and Ehrnsten et al. (2020b).**

Further, empirical studies of faunal effects often focus on temporally and spatially limited parts of the system, omitting important interactions and variability occurring in natural ecosystems (Snelgrove et al., 2014). **It is logistically challenging to study multiple drivers and interactions in the benthic and pelagic realms, such as the interactions between benthic and pelagic production, empirically. Mechanistic or process-based models are powerful tools to conduct such studies (Seidl, 2017).** Here, we extend a physical–biogeochemical model of the Baltic Sea ecosystem (BALTSEM; Gustafsson et al., 2014; Savchuk et al., 2012) with benthic fauna components based on the Benthic Macrofauna Model (BMM; Ehrnsten et al., 2020a). We include both the direct feedbacks from animal growth and metabolism and the indirect effects of their bioturbating activities on biogeochemical cycling to evaluate their relative contributions.

We use the Baltic Sea as a model area for **three** reasons: (i) the shallow depth (mean depth 57 m) and enclosed geography with a long water residence time (about 33 years) contribute to strong benthic-pelagic coupling (Snoeijs-Leijonmalm et al., 2017; Stigebrandt and Gustafsson, 2003), **thus pelagic nutrient dynamics are highly dependent on benthic processes, (ii) the relatively simple, species-poor benthic communities facilitate model development,** and (iii) the major features of biogeochemical cycling of C, N and P in the Baltic Sea are well known due to a wealth of oceanographic measurements and studies performed over the past century, making it an ideal system for process-based modelling (Eilola et al., 2011; Gustafsson et al., 2017; Savchuk and Wulff, 2009, 2001). However, the sediment pools and the role of sediment processes in benthic–pelagic exchange are not as well quantified as pelagic pools and fluxes. **The higher uncertainty in benthic compared to pelagic processes as well as the traditional focus on pelagic eutrophication are probable reason why physical-biogeochemical models of the Baltic Sea have omitted benthic fauna as state variables (e.g. Eilola et al., 2011; Lessin et al., 2018).** Here, we aim to fill this knowledge gap and explore the role of benthic fauna in biogeochemical cycling of C, N and P on a long–term ecosystem–level scale."

**Materials and methods**

1.    Line 74: the focus of the study is "on the Baltic Proper and the Gulf of Riga". Could you please explain your choice of focus areas in more detail? While the focus on Baltic Proper as the largest and deepest basin is straightforward, why Gulf of Riga? It is the only sub-basin to the east of the Danish Straits for which there was no validation data available apart from some literature-based values, so it is not possible to adequately validate the model for this region, and consecutively the confidence in model performance here is lower than in other areas, both in terms of absolute biomasses of benthic fauna and its impact on pelagic biogeochemistry.

We realize we did not properly motivate the choice of focus, which we hope is now clarified with the following addition (lines 85-88):

"In this study, we focus on comparing results from the Baltic Proper and the Gulf of Riga (Fig. 1), two basins with a similar benthic community composition but differing in physical and biogeochemical properties such as depth, openness, productivity and bottom oxygen conditions. We expect these differences to be reflected in the strength of benthic-pelagic coupling processes and the role of benthic fauna therein."

Further, we found new data to validate benthic fauna biomasses in the Gulf of Riga (93 data points in the appendix of Gogina et al. (2016)). This new validation analysis conforms with the previous comparison to literature values: the simulated mean total biomass is higher that observed (mostly due to an overestimation of *M. balthica*), but the simulated biomasses of all groups are well within the standard deviations of data. We also added several new estimates of benthic fauna biomass in the gulf found in the literature to Table 1. The new estimates from the 1980s are a fairly close match to simulations. Please see our response to RC2, Line 221 for details. Corresponding changes to the results text were made on lines 244, 252; discussion on lines 341-347; Figure 3c; Appendix C on lines 972-973, 981-988; Table C1 and Figures C1-C2.

2.    Figure 1: As the text often refers to different regions of the Baltic Sea, a legend for all of them should be provided in addition to numbers. Moreover, model results are presented at intervals of 0-30, 30-70, 70-120 (and 120+) meters, so bathymetry on the figure should use the same gradation.

We thank the reviewer for the suggestions, which will significantly aid the reader. We have changed the figure accordingly (Figure 1).

3.    Section 2.2: BALTSEM simulates Baltic Sea as 13 horizontally homogeneous boxes. In the Results and/or Discussion section, it should be elaborated on how this affects model results. Some of the Baltic Sea sub-basins have strong gradients in nutrient distributions, which in turn leads to gradients in productivity and in distributions of benthic fauna. For instance, the eastern part of the Gulf of Finland is heavily influenced by riverine nutrient inputs and have higher primary production rates than its western part. How are these gradients accounted for?

The horizontally averaged approach does mean that finer scale gradients in e.g. nutrient distributions cannot be accounted for. This is why we have excluded data from archipelago areas, as the model does not represent the small-scale complexity in these areas (as explained in the Methods section of Appendix C).

Although horizontal integration indeed is a compromise, comparisons with horizontally resolved models showed that BALTSEM is as good as these in reproducing seasonal and long-term variations, at least outside hot-spot areas like the eastern Gulf of Finland and major river mouth areas (Eilola et al., 2011; Meier et al., 2018).

4.  Lines 128-130: why does the model consider respiration and excretion of benthic fauna to contribute *ammonia* and phosphate directly to the water column? At least in case of deposit feeders one might expect part of the excreted ammonia to be oxidised directly within sediments and to be released into the water column in form of nitrate, and some of the excreted phosphate to be bound within sediment, the same way as it is considered for microbially remineralised nutrients in the model?

It is true that part of animal excretion takes place within the sediment and is oxidized before release to the pelagic. However, we do not have good estimates on what that proportion could be. Given that there are no head-down/subsurface deposit-feeders in the Baltic Sea *sensu stricto*, and the main species are either mobile surface dwellers (e.g. *Saduria entomon, Monoporeia affinis*) or infauna living with head or siphons at the surface (e.g. *Macoma balthica, Marenzelleria* spp.), we can assume that the majority of excretion is released to the water column. Following the principle of Occam's razor, we try to keep the model as simple and interpretable/traceable as possible, and therefore refrain from dividing the excretion fluxes. In oxic conditions (which almost always prevail in areas with benthos), the excreted ammonia is immediately oxidized in the water column.

5.  Does the model consider sediment resuspension? It seems to be an important factor mediating organic matter availability in the sediments in coastal seas, especially shallow regions such as Gulf of Riga? Omitting it might have important consequences both for model parameterisation and its results?

BALTSEM includes downward relocation of sediments due to resuspension and lateral transport. A description of the process can be found in Savchuk et al. (2012) and in the Supplementary material of Ehrnsten et al. (2020a).

6.  Line 134: "degradation" - what type of degradation? At this point, it should be specified in more concrete terms.

We changed the term "degradation" to the more specific term "mineralization" (line 148).

7.  Around line 152: how are silicate transformations handled in the benthos?

We assume that Si does not interact with the fauna (i.e. storage or transformation of Si by fauna is not important for model dynamics), and therefore kept the original formulations for sediment Si from BALTSEM, i.e. it is treated as single pool with sinking diatom Si as a source and mineralization and burial as sinks (see Appendix A, Eq. A4). We added a short mention of this in the main manuscript (lines 148-150):

"Sediment C, N and P pools are further divided into three banks of different age to resolve the food limitation of benthic fauna (Fig. 2), while benthic Si is represented as a single pool that does not interact with the fauna."

8.  Line 167: is nitrification considered a sink for oxygen in the model?

Yes, nitrification consumes oxygen according to Eq. A17 in Appendix A. We added a mention of this to the sentence (lines 178-179).

9.    Equation 8, page 7: is sequestered phosphorus considered a state variable in the model, as it is not listed among the state variables in Table A1?

Sequestered P is not separated as a state variable, but is instead returned to the oldest sediment bank SED3P (see Figure 2). We will not endeavour into speculation about how a separate state variable would affect results of the current study, but we are looking forward to see the results of an ongoing study implementing BALTSEM with sediment iron and iron-bound P as explicit state variables (unfortunately without benthic fauna, though).

10.    Line 176: what is a definition of "severely hypoxic"?

The oxygen concentration value when P sequestration switches to release depends on salinity and bioturbation, and therefore it is difficult to give an exact value. At salinity 0 and Ebio=0.6, the value is 0 mg $O_2$ $L^{-1}$, while at salinity >5 and Ebio=0 it is 1.44 mg $O_2$ $L^{-1}$. Please see our response to RC3 for graphs of the oxygen-dependency of benthic N and P processing.

11.    Section 2.3: please provide more detail on model forcing and setup in addition to the references, so the interested reader does not need to look for those details within several previous papers. How was benthic fauna initialised?

We added a short description of model forcing and initial conditions on lines 202-211. For the interested reader, we refer to the given references for further details.

"The model was run over 1970–2020 forced with observed nutrient loads and actual weather conditions as described in Gustafsson et al. (2012, 2017) with forcing time–series extended to 2020. The physical circulation was forced by 3-hourly meteorological conditions and monthly time-series of river runoff and state variable concentrations and sea level at the North Sea boundary. Monthly inputs of N, P, C and Si from land via rivers and from coastal point sources as well as atmospheric deposition of N, P and C were used as biogeochemical forcing. Initial conditions in 1970 were based on observations for pelagic variables and hindcast simulations for benthic variables as described in Gustafsson et al. (2012) and Ehrnsten et al. (2020a). Shortly, the benthic fauna and their food banks (SED1X and SED2X) were set to 1000 mg C $m^{-2}$, 100 mg N $m^{-2}$ or 10 mg P $m^{-2}$ throughout the model domain in 1960 and given 10 years of hindcast simulation to spin up, allowing the variables to turn over several times. Initial conditions for SED3X, with a slower turnover rate, were based on a hindcast from 1850 to properly account for the build-up of sediment nutrient pools during past eutrophication (Gustafsson et al., 2012)."

12.    Line 184-185. Aggregating the results to means and standard deviations obscures a lot of detail about model capabilities, which cannot be justified by stating that "the purpose of this study was to evaluate large-scale dynamics". Does the model capture seasonal dynamics of benthic fauna? Are there long-term trends in the model *or* in the data? There is a general lack of knowledge about benthic processes, so more detailed and varied comparison with data would allow to identify critical gaps and steer discussions which will help to identify directions for improvement. This is especially important for a new model implementation, as presented in the manuscript.

The main aim of this study is to examine the effects of benthic fauna on biogeochemical cycling of C, N and P on a system scale. For this purpose, we believe that aggregation of results into long-term (20 year) means and standard deviations is adequate. Before we can get to the main aim, we need to validate the model, and we believe that this is the part the reviewer is primarily referring to in this comment.

As much as we would like to add validation of further benthic variables and processes, unfortunately there no data available for further validation. While there probably is a large amount of data on e.g. sediment concentrations and fluxes measured for research and monitoring purposes, to our knowledge there is no comprehensive collection of the data available in the form of an open-access database or review article. As we comment also in response to RC2 and RC3, we restrict the validation to these types of transparent, open-access sources and believe it would be far beyond the scope of this study to do a comprehensive collection of data from unpublished sources or literature. Thus, we restrict the quantitative validation to pelagic variables (Appendix B: salinity, temperature, oxygen, ammonium, nitrate and phosphate), benthic fauna (Appendix C) and sediment C:N:P ratios (results section 3.1). In addition, we put our results into context of previous research, including quantitative and qualitative evaluations of sediment processes and the role of fauna therein, in the Discussion.

The dynamics of the two component models has been extensively validated before, including seasonal and long-term dynamics in BALTSEM (Eilola et al., 2011; Gustafsson et al., 2012, 2014; Meier et al., 2018; Savchuk et al., 2012), as well as long-term dynamics of the Benthic Fauna Model (Ehrnsten et al., 2019, 2020a). We are not aware of any data for validation of seasonal dynamics of benthic fauna, but we also believe that the relevant time-scale of benthic processes in a system perspective is years rather than seasons. Thus, we would not like to add further comparisons of temporal dynamics to this manuscript, since (1) the manuscript is already 52 pages long, (2) many aspects have been validated previously, and (3) to our knowledge, there is no data available for further validation of benthic processes.

13.    Lines 187-188. This is an important topic for benthic modelling and should be discussed in some more detail. What should observational scientists measure to help constrain the model?

What would be needed first and foremost is a comprehensive collection of observational data from the benthic realm in a format that can be used for validation of basin-scale stocks and fluxes, similar to existing databases for pelagic variables. We have added a section discussing this to the conclusions and outlook on lines 435-468.

**Results**

1.    The model run is 1970-2020. Why was validation of pelagic state variables performed for 1970-2015?

2015 is the latest year we have a full compilation of validation data for, therefore the validation only extends to 2015.

2.    Figure 8 shows that there was a noticeable impact of inclusion of benthic fauna/bioturbation on primary productivity and nitrogen fixation, so even if overall relative bias remained almost unchanged, there was an impact on pelagic dynamics. Please provide more detail on how the models with BMM and without BMM compare. Given the

impacts of fauna and its activity on nutrient fluxes is the main scope of the paper, a more detailed assessment of changes in pelagic environment is justified.

As the model with BMM has been recalibrated compared to model without BMM, it would not be correct to compare the two model versions directly to infer the effects of fauna. We realize that this was not properly explained in the manuscript, and have now included a section on the recalibration process in the Materials and Methods section (lines 195-200). Shortly, BALTSEM without BMM uses a simple first-order formulation for mineralization of sediment stocks. This implicitly includes mineralization by microbes and benthic fauna. When the benthic fauna was added as explicit variables that perform mineralization through metabolic processes, the first-order mineralization rates ($a_{ZSEDC}$, $a_{ZSEDN}$ and $a_{ZSEDC}$ in Table A2) were decreased to avoid "double-counting". Instead of comparing model versions, we use the sensitivity analyses, scenarios and comparison of the two contrasting basins to infer the effects of fauna under various settings.

The recalibration also explains why the overall relative bias is unchanged: we aimed for producing a model with added dynamics that works as least as well as the previous version of BALTSEM, which has been developed over several decades to reproduce the main biogeochemical cycling processes in the Baltic Sea. Thus, we did not aim primarily at improving model performance in this sense, but rather to create a tool for exploring the effects of benthic fauna on biogeochemical dynamics. In fact, it took us about two years to get to the stage where the coupled model performs about equally well as the uncoupled model.

Regarding the second part of the comment, we agree that more detail on the effects of fauna on the pelagic processes is warranted, and have added text and figures on changes in primary production, nitrogen fixation, sedimentation and oxygen conditions to the results and discussion as suggested here and in several other comments. Additions and changes are found on lines 279-281, 284-299, 308-318, 438-449 and Figures 8 and 10.

3.    Line 214: how is "reasonable accuracy" defined in case of benthic variables? As the authors demonstrate, benthic fauna biomass shows high variability, hence large standard deviations, especially when aggregated over long time and large areas, while standard deviations in the modelled fauna are relatively small. Does this high std in data really justify using cost function (Appendix C) as a validation metric?

"Reasonable accuracy" is defined according to Eilola et al. (2011): model results can be interpreted as good if the model mean is within one standard deviation of the observed mean (0≤CF<1), reasonable if 1≤CF<2 and poor if CF≥2. This was explained in Appendix C, but is now also defined in the main text (line 245-246).

High variability in benthic fauna data is inherent to variability that is not expected to be found the model results (e.g. due to patchiness of habitats and resources), so therefore we think it is reasonable to investigate whether the model mean is within the variation of the data. We are open with the fact that large standard deviations in data give some "slack" to the Cost Function estimates (discussed both in the main manuscript and Appendix C), but we still think that Figure C1 gives a good overview of the model performance with respect to benthic fauna. All data behind the cost function is shown in Figure C2 to allow the reader to judge the performance of the model in greater detail.

4.    Line 221-224: see my comment above: why focus on the Gulf of Riga?

Please see our response to comment 1. (Materials & methods) above.

5.    Section 3.3, from line 247: this is a key section in the manuscript, as it demonstrates the direct and indirect effects of fauna. Yet it is very short and just skims through the results and through the figures. It should be more detailed, which would also make it easier to relate figures 7-9 to the text. It would also benefit from detailed quantitative information, in particular regarding relative changes.

We have added more detail to the existing text (especially % change), and also added new paragraphs delving more into the pelagic dynamics. As suggested further down, we added result graphs on the extent of hypoxic and anoxic areas under different levels of bioturbation to Figure 8. Please see extended section 3.3 on lines 271-299 and response to comment 2. above.

6.    Line 264: for this, a figure similar to figure 5 could be provided.

We could add a figure on the depth distribution of fauna and bioturbation coefficients, as suggested. However, we already have 11 figures with multiple panels in the main manuscript plus several more in the appendices. We are not sure that this figure would add enough new information compared to Fig. 10 that gives the same information aggregated over depth. We leave it up the reviewer's and/or editor's discretion to judge if a figure similar to one below should be included. In that case, we can rerun the model with higher depth resolution output to get smoother curves.

[Figure]

**Possible new figure.** Depth distribution of benthic fauna biomass and the bioturbation coefficient $E_{bio}$ in the upper 100 m of the Baltic Proper and the Gulf of Riga. Averages (lines) and standard deviations (shaded areas) of biweekly values 2080-2100 in the BSAP (a-b) and HIGH nutrient load scenarios (c-d).

7. Lines 265-271: figure 11 (as well as several other figures of the manuscript) contain multiple subfigures, so it would be very helpful to provide relevant pointers in the text, e.g., reference to Fig 11a on line 268, Fig 11b on line 269 etc, so the text and figures could complement each other.

We have added references to figures and subfigures in results and discussion as suggested (lines 268, 280-281, 285-299, 304-305, 309-318, 336, 340, 342, 350, 360, 360-375, 385, 415,427 and 444).

**Discussion**

1. Line 284: both terms "long-term" and "large-scale" have been used multiple times in the manuscript without proper definition. It creates overall impression of vagueness. Does "long-term" stand for "multidecadal" or "long-term mean"? The words "large-scale" could be omitted altogether without impact on meaning.

We have used the terminology of BALTSEM, which stands for "Baltic Sea Long-Term Large-Scale Eutrophication Model". In this context, the former refers to the decadal time-scale of simulations (i.e. long-term mean), and the latter to (a) the organisational scale of the ecosystem and (b) the spatial scale of basins. We agree that the terms were used in a slightly vague sense in many places and have removed or replaced the terms with more specific definitions where applicable (lines 1, 23, 214 and 448). We also removed the term "Large-scale" from the title (see first comment above).

2.   Line 285: as it currently stands, it should be "Baltic Sea", not "coastal sea", as the manuscript shows the model is not yet applicable in high-salinity regions.

This is a good and valid point, we changed the expression as suggested (line 327).

3.   Line 288-289: to support the statement that benthic fauna can alleviate the 'vicious circle' of eutrophication, could you show the differences in extent of hypoxic area (e.g. with Figure 8) for simulations with different levels of bioturbation?

We thank the reviewer for the good suggestion. We added graphs showing the extent of hypoxic and anoxic areas under different levels of bioturbation to Figure 8, as well as a section to results giving more detail regarding changes in oxygen conditions, N fixation and primary production (lines 284-299).

4.   Similarly, what about primary production and N-fixation for the two future scenarios (Fig 10)? I think these would also support the discussion on faunal impacts on 'vicious circle'?

We added graphs on primary production and N-fixation in the load scenarios to Figure 10 as suggested (referenced on lines 304 and 322). These graphs nicely illustrate the interesting result that the relationship between primary production and sedimentation changes with nutrient loads, as discussed in section 4.3.

5.    From line 307: as the overall model performance was not improved by adding benthic fauna, and related processes are not well understood, could the authors discuss on what understanding is currently lacking and which model improvements are desired.

Please see our response to comment 2. in results above.

6.   Lines 313-316: same as the previous comment, which data is currently lacking? Which mechanisms are not sufficiently understood? The sentence on "the main strength of this study" is too general ("many interlinkages") – could it be replaced with something more concrete and relevant to the actual modelling work?

We realise that this paragraph was badly formulated, as also pointed out by Reviewer 3. It is now removed from the manuscript. Instead, we have expanded section "5. Conclusion and outlook" with a discussion on challenges, opportunities and ways forward (lines 432-468).

7.   Lines 326-334: please keep referencing to figures and sub-figures as discussion requires.

We have added references to figures and sub-figures as suggested (lines 268, 280-281, 285-299, 304-305, 309-318, 336, 340, 342, 350, 360, 360-375, 385, 415,427 and 444).

8.    Line 361: is it worth presenting fast running time as an advantage, since the manuscript presents results from only several simulations with sensitivity analysis limited to a single parameter?

While the manuscript presents results from just a few simulations, the work behind the manuscript comprises hundreds if not thousands of simulations for different purposes including integration of code from the two models, debugging, parameterization, calibration etc. We doubt that we would have been able to implemented the model integration and analyses as successfully if we would have used a complex 3D-model with a running time counted in days rather than minutes. We have added a paragraph to Materials and methods explaining the calibration process (lines 195-200).

**Conclusions and outlook**

1.    Line 397: "much-studied bioturbation" - I suggest replacing this with "bioturbation, relatively more studied in the modelling context".

Changed as suggested (lines 435-436).

2.    Could the authors provide some outlook on the potential future directions of benthic modelling, in the context of Baltic Sea in particular?

3.    Lines 399-402: These conclusions are far too general. Could it be something emerging from the study rather than vague statement that benthic-pelagic coupling is "modified by multiple drivers, which may change over time"?

4.    Based on your work, has the time arrived for the regional models (of the Baltic Sea) to extend their formulations to explicitly include benthic fauna?

Response to comments 2.-4.

We rewrote the section to (a) be more specific about the findings of this study, and (b) provide an outlook for future directions of benthic modelling in the context of physical-biogeochemical models of the Baltic Sea and beyond. The new version is found on lines 433-468.

Appendix A, line 817: why is mortality rate chosen to be linear for *Limecola balthica*, and quadratic for the other two groups?

Quadratic mortality rate is generally used as a closure term to represent predation by groups not present in the model. This is the reasoning behind using a quadratic term for the benthic predator group and linear terms for its two prey groups, the deposit-feeders and *M. balthica* in the first version of the fauna model (Ehrnsten et al., 2019). As the model was extended to the Baltic Sea scale, the mortality rate of deposit-feeders was changed to quadratic to compensate for missing predation pressure in some areas with low or absent biomass of predators (primarily in the oligotrophic Bothnian Bay, Ehrnsten et al., 2020a).

**Technical corrections**

1.    Line 89: "extension" - should be "extent" Corrected as suggested (line 103).

2.  Line 110: "Ponotporeia" - should be "Pontoporeia" Corrected as suggested (line 124).

3.  In equation 1, should "Uci" be "UBFiC" in the denominator? Corrected as suggested (Line 158).

4.  Page 6: equation 6 is missing. The mistake in equation numbering was corrected. Equations are now numbered consecutively from 1 to 9 (lines 182-194).

5.  Line 175 and equation 10: the fitting constants 5-8 are not featuring in the equation. The extra fitting constants were removed (line 190). The terminology was also changed (see response to RC3).

6.  Figures 7 and 11: in both cases, "phosphorus (c, d)" should be "phosphorus (e, f)". Corrected as suggested (Figs 7 and 11).

7.  Line 319: "suggests" - should be "suggest" Corrected as suggested (line 360).

8.  Table 1: what is "total" biomass? Should it be "mean" instead? Corrected as suggested (Table 1).

9.  Line 712: "and" - should probably be "an" Corrected as suggested (line 788).

10.  Line 742: Nitrite and nitrate should be "NO2" and "NO3", respectively. Corrected as suggested (line 818).

11.  Line 784: "and" at the end of the line should be "are"? Corrected as suggested (line 861).

12.  Line 794: "feces" - "faeces". Corrected as suggested (line 870).

13.  Line 907: "120-120" should be "70-120" Corrected as suggested (line 987).

14.  Figure C2: please use a different y-scale for the Bothnian Bay. Scale changed and a note about differing scale added to figure legend (Figure C2).

References in review response

Ehrnsten, E., Norkko, A., Timmermann, K. and Gustafsson, B. G.: Benthic-pelagic coupling in coastal seas – Modelling macrofaunal biomass and carbon processing in response to organic matter supply, J. Mar. Syst., 196, 36–47, doi:10.1016/j.jmarsys.2019.04.003, 2019.

[revised manuscript text omitted]

Stigebrandt, A. and Gustafsson, B. G.: Response of the Baltic Sea to climate change - Theory and observations, J. Sea Res., 49(4), 243–256, doi:10.1016/S1385-1101(03)00021-2, 2003.

**Response to RC2 by Hans Cederwall**

Please see our response to each comment below in blue. Revised line numbers and figures are added in orange.

This is a good paper trying to merge a benthic model with the older pelagic model by Savchuk and Wulff. The result is interesting, furthering modeling work is encouraged.

Remarks

Line 3, 22 and other places: Bioturbation in strictu means the mixing and turning over of the sediment. In the Baltic Proper that is dominantly carried out by the two amphipod species, who lies borrowed in the sediment during daytime, but swims about searching for food on the sediment surface during nighttime (they normally bioturbate the uppermost cm, down to ca 5 cm). To some part also Saduria takes place in the bioturbation but only on the suface layer. Species like Macoma, Halicryptus and Marenzelleria who are more or less permanently burrowed in the sediment (often deeper than 5 cm). They are more of bioirrigators. It should be mentioned early in the paper that the authours in their term bioturbation includes bioirrigation.

It is true that the definition of bioturbation often excludes bio-irrigation. We have chosen to use the definition of Kristensen et al. (2012), who proposed that for consistency, bioturbation (by fauna) should be defined as an umbrella term that covers "all transport processes carried out by animals that directly or indirectly affect sediment matrices. These processes include both particle reworking and burrow ventilation." We added this definition to the introduction (lines 39-40):

"Here, we define bioturbation as all biological processes that affect the sediment matrix, including burrow ventilation (bio-irrigation) and reworking of particles (Kristensen et al., 2012)."

Line 65: Here is stated that the penetration depth of oxygen in Baltic Sea sediment is usually measured in mm rather than cm. This is only true for deeper bottoms from ca 60m or deeper. It could of course be found at shallower depth i archipelago areas, but those areas has been excluded in this work. I have measured redoxpotential at zoobenthos stations for more than 30 years and only on deep stations found oxygenation only in the top mm. The model here includes a depth interval of 0-120 m, so in the uppermost 60 m oxygenation is definitely better than what is said here.

We agree that the refences given (Almroth-Rosell et al., 2015; Bonaglia et al., 2019; Hermans et al., 2019) are biased towards deeper parts of the Baltic Sea, and may not give a representative picture. We therefore removed the statement.

Line 108: Limecola balthica has changed back to Macoma balthica (Caroline Raymond, pers. comm.)

We have happily changed the species' name to the newly (re)accepted *Macoma balthica* throughout the manuscript, following the update in the World Register of Marine Species: https://www.marinespecies.org/aphia.php?p=taxdetails&id=141579

Line 110: Pontoporeia is misspelled

Spelling corrected (line 124).

Line 124: Misspellings: basalmaintenance and biomas s

These errors were due to a problem with the pdf conversion, and will be corrected in the next version of the manuscript.

Line 183: Why hindcast simulations for benthic variables.? There was a lot of data collected in the 1970s. For example in the Joint sampling programme (Elmgren 1978) and by revisiting the Hessle-stations (Cederwall & Elmgren 1980). There should also be a lot of finnish data collected in the 1970s by Lassig and Andersin from the Finnish Institute of Marine Research. As far as I know these data were transferred to SYKE when the FIMR was closed down. Data from the Joint Sampling Programme and the Hessle project is stored in the benthic database BEDA. Contanct Caroline Raymond or Mats Blomqvist.

The model needs 1349 initial conditions for each benthic variable, one for each depth meter in each basin. While it would potentially be possible to use data collected around the 1970s to initialize the model, it would include a lot of extrapolation and guesswork and would probably not cause any significant changes to the main results of the current study focusing on the period 2000-2020, as the variables have time to turn over several times from the start in 1970. However, we consider this a very interesting proposition and will look into the availability of historical data and possibility to compile and use them for future studies.

Line 213: I strongly suggest you leave at least the Arkona Basin out, beause the fauna here contains several other species than the ones you have mentioned in this preprint. Stick to the Gotland Basins, Gulf of Finland and Gulf of Riga who has a similar set of species.

We agree that the model is best applicable from the Bornholm Basin and north, where the composition of fauna is similar. We moved the biomass comparison for the Arkona Basin from the main manuscript (Fig 3) to Appendix C and changed the sentence to (lines 246-248): "Simulated mean biomasses of the individual functional groups and the groups combined were mostly within one standard deviation of observed means from the Bornholm Basin (basin 8) in the south to the Bothnian Bay (basin 11) in the north, although it should be noted that the spread of observed data is large."

We also changed a sentence in the Appendix to reflect the limited applicability of the model in Arkona (lines 995-996): "The model is not applicable to the high-salinity areas at the entrance to the Baltic Sea (Kattegat, Öresund and Arkona Basin), as it does not include the high diversity of functional groups present in these areas (Figs. C1–C2)."

Line 221, and Table 1: The benthic database BEDA contains primary data for 7 stations sampled in 1976 for the Joint Sampling Programme. In the mid 1990s a mapping of the macrofauna of the Gulf was done within the NMR financed Gulf of Riga Project. The results are published (Cederwall et al. 1999). Possibly the primay data are still held by Vadims Jermakovs, Latvian Institute of Aquatic Ecology (An institute where your colleague Bärbel Muller-Karulis earlier worked).

We thank Dr. Cederwall for the suggestions for additional resources on benthic fauna biomasses. We also acknowledge that the SMHI Sharkweb database relies on quality checks by the data deliverers and has a very limited quality check within itself. However, we would still argue that our approach using only published and/or open-access data is a reasonable choice. The main purpose of the biomass validation in this study is to confirm that the validation performed by Ehrnsten et al. (2020) still remains valid in the modified model version. Therefore, we believe it would be beyond the scope of the current study to make a new, comprehensive collection of data from unpublished sources. We have made extensive quality checks of the data from both Finnish and Swedish databases to reduce the error introduced by data quality issues, as explained in Ehrnsten et al. (2020) and in Appendix C.

To improve model validation for the Gulf of Riga, we now extended the validation with data from Gogina et al. (2016). In their appendix, data on wet biomasses for benthic species is provided. The data is based on a comprehensive compilation from several sources up to 2013 and is provided as means for 5 km$^2$ squares. Within the Gulf of Riga excluding the shallow coastal areas, data for 95 squares were used. We classified and analysed this data in the same way as the other validation data and now include the Gulf of Riga in the quantitative validation in Figs. 3, C1 and C2. Even though the data is slightly more aggregated than the previously used data sources, we believe this gives a good picture of the general biomass range in the Gulf together with the comparison in Table 1 (extended with two publications, see response under Table 1 below). Conforming with the original manuscript, the new analysis shows that simulated mean total biomass was higher than observed. With the new data, we can see that this is mostly due to higher simulated biomass of the dominating group, *Macoma balthica*. Possible reasons for overestimation are already mentioned in the original manuscript section 4.1 (lines 299-303). The new cost function values vary between 0.11 and 0.75, i.e. simulated biomasses were well within one standard deviation from data. Simulated biomasses are also within the range of the new data in Table 1.

Corresponding changes to the results text were made on lines 244, 252; discussion on lines 341-347; Figure 3c; Table 1, Appendix C on lines 972-973, 981-988; Table C1 and Figures C1-C2.

Line 277-279: Is the big difference in sedimentation between the BSAP scenario and the HIGH load scenario reflecting a difference in phytoplankton species composition? Historically the main input to benthic ecosystem has been the sedimentation of diatoms during the spring bloom, not the sedimentation during blue-green blooms. There has however been a shift in species composition in the spring bloom, where diatoms have decreased and other groups (who have lower sinking rate) have increased (Hjerne et al. 2019).

Indeed, the difference in sedimentation in relation to primary production rates reflects differences in timing and composition of primary production. BALTSEM simulates three types of phytoplankton: diatoms, N-fixers (i.e. cyanobacteria) and 'other species', representing mainly summer-blooming flagellates. Diatoms have the fastest and cyanobacteria have the slowest sinking rate. In addition to sinking rate, the proportion of primary production reaching the seafloor depends on zooplankton grazing rates, bacterial remineralisation rates and physical properties of the water column (e.g. stratification). Thus, sedimentation is determined by a suite of interacting processes, and the relative contribution of each is not straightforward to tease out. In general, though, BALTSEM simulates a shift from spring to summer blooms and increase in cyanobacteria in response to eutrophication and warming over the past decades, as briefly discussed in section 4.3.

We have added graphs on primary production and N-fixation in the different scenarios to Fig. 10. These show that the rate of N-fixation increases strongly with nutrient loading, implying a change in phytoplankton composition. N-fixation is about 5 times higher in the HIGH load scenario than in the BSAP scenario in the Baltic Proper, while N-fixation is completely absent in the BSAP scenario in the Gulf of Riga.

Line 397: cf Mäkelin & Villnäs 2022. Could the seasonal variations in benthic stoichiometry have any influence on your modelling work?

General theory predicts that animals regulate their stoichiometry within a narrow range (Sterner and Elser, 2002), which is the basis for our assumption of a fixed stoichiometry in the benthos. However, as shown for example in this very interesting study, some variation does exist. We are grateful to our colleagues Mäkelin and Villnäs for sharing their preliminary results with us, which were used to parameterise the model. We tested different stoichiometric ratios based on the range

found in the study. Indeed, by changing the C:N weight ratio in the benthic fauna from 6 to 7, the growth of fauna shifted from being mostly N-limited to mostly C-limited. However, the quantitative difference in biomass was relatively small. In other words, a variable stoichiometry could have an impact on growth and excretion estimates, but we would need more information on how, and especially why, the stoichiometry varies over time before implementation in this ecosystem-scale model would be feasible.

Fig 3: You show bars for the depth interval 0-30 m. I strongly suspect you have very few if any stations in the depth interval 0-10 m. This because you have outruled data from archipelagos and open sea research vessels don't like to go into shallow waters. Also these areas are dominated by transport bottoms hard to sample quantitately. Finally in these shallow bottoms you have another set of species than the set you have worked with. You should change to the depthinterval 10-30 m. In the text you mention that you had no shallower (<30 m) data from the Arkona Basin. The benthos database BEDA contains some data from the 1980s (Mats Blomqvist, pers. comm.) There is also a lot of data held by German institutes. But on the other hand I advise you to leave the Arkona Basin out of your paper since the fauna there differs so much from the Baltic Proper.

As suggested, we removed the Arkona Basin from Fig. 3 and modified the text as explained in response to Line 213 above.

While it is true that we have excluded a large part of the shallow areas with a diversity of habitats and communities, we still included data from depth interval 0-10 m, making up ca 11% of the data in the <30 m depth category. We could exclude this data from the comparison, but we believe it is more correct to include it since we cannot exclude this area from the model.

Table 1: Could becompleted with Cederwall et al. 1999.

This excellent publication was surprisingly hard to find, but we eventually got hold of it through the Technical Library of Hamburg. We added the data to Table 1. We also added data from another publication to the table (Gaumiga and Lagzdins, 1995). The data in these publications seem to strengthen the validation of the model: simulated biomasses are well within the range of the reported observations (lines 344-345).

Table A4, footnote: Mäkelin & Villnäs is published in Limnol. Oceanogr. 2022. The reference is missing in the reference list.

Reference to the recently published study was added as suggested (Table 4 footnote, reference list).

Line 901: You should know that sharkweb/SMHI does not have a benthological quality control of the data delivered to them. They assume that laboratories deliver correct data. This is not always the case (Mats Blomqvist pers. comm.). I suggest you import data from BEDA.

Please see our response to comment to Line 221 above.

Final comment: The macrofauna is not the only part of the benthic ecosystem. The meiofauan for instance can have biomasses of 5-10 g wetweight/m2 (Elmgren 1976). To what extent does the meiofauna influence the sediment chemistry?

It is possible that the meiofauna is important for sediment processes, but since it is an understudied group compared to macrofauna, we do not have the proper means to estimate their effects. In a modeling context, it is also a bit tricky to include the meiofauna as a functional group, as the grouping is based on size rather than function.

**Summing up**: This is basically a good paper well worth publishing after revision.

References in the response

Blackford, J. C.: An analysis of benthic biological dynamics in a North Sea ecosystem model, J. Sea Res., 38(3–4), 213–230, doi:10.1016/S1385-1101(97)00044-0, 1997.

Gaumiga, R. and Lagzdins, G.: Macrozoobenthos, in Ecosystem of the Gulf of Riga between 1920 and 1990, edited by H. Ojaveer, pp. 196–211, Estonian Academy Publishers., 1995.

Gogina, M., Nygård, H., Blomqvist, M., Daunys, D., Josefson, A. B., Kotta, J., Maximov, A., Warzocha, J., Yermakov, V., Gräwe, U. and Zettler, M. L.: The Baltic Sea scale inventory of benthic faunal communities, ICES J. Mar. Sci., 73(4), 1196–1213, doi:10.1093/icesjms/fsv265, 2016.

Kristensen, E., Penha-Lopes, G., Delefosse, M., Valdemarsen, T., Quintana, C. O. and Banta, G. T.: What is bioturbation? the need for a precise definition for fauna in aquatic sciences, Mar. Ecol. Prog. Ser., 446, 285–302, doi:10.3354/meps09506, 2012.

**Response to anonymous reviewer comments RC3**

Please see our responses in blue below. Revised line numbers and figures are added in orange.

In this paper, a new tool is presented that couples a low-resolution pelagic biogeochemical model with a low resolution benthic biological model for the Baltic sea. The (vertically integrated) benthic state variables are then used to calculate impacts on biogeochemistry using presumed effects of bioturbation and water-column conditions on denitrification and phosphorus dynamics. My main doubts with this paper are connected to the biological focus of the model.

Essentially there exist two schools of modelers: some modelers take a *biological* approach and ignore or strongly parameterize biogeochemistry. Their models disregard the small-scale vertical gradients of solutes in the sediment and often consider only surface-averaged concentrations of particulate substances (e.g. organic matter). Moreover, their models operate on seasonal time scales, as organisms usually react on these time scales.  Opposed to this are the modelers that tackle sediment dynamics from a *biogeochemical* perspective and strongly parameterize biology. These modelers take into account the fine-scaled vertical gradients of solids and solutes that are observed in the sediment, and their dynamics includes reactions operating at very different timescales, from very short (< seconds) up to very long time scales (multi-years). In these models, the metabolism of the (higher) organisms is included as "oxic mineralization" of organic matter, while their bioturbation activity is included as a "coefficient". Thus, these models strongly parameterize the biology, and only explicitly account for the biogeochemistry. As long as the main conclusions of these models are stated in the area of the model focus, there is nothing wrong with any of these approaches. For instance, it is reasonable to assume that a biogeochemical model can rather faithfully reproduce the impacts of certain external conditions on sedimentary nitrogen or phosphorus removal rates, but it is questionable whether such models can also well represent the distribution of the benthic organisms that drive the biogeochemical cycles. Similarly, why would we put a lot of faith in biogeochemical conclusions that come from a model that focusses on biology and parameterizes the biogeochemistry? This is in a nutshell the doubts I have on this paper. While the conclusions seem logical, I am still to be convinced that the tool used to arrive at them is appropriate.

Because of the biological focus, there are quite some assumptions with respect to biogeochemistry that are not dealt with in the manuscript. For instance: the paper talks about the sediment pools of C, N and P, and Si. Biogeochemically one distinguishes between particulate and dissolved pools – here I had to guess that the pools refer only to particles (the 'food' of the organisms). Thus, the transient (within season) storage of dissolved components is ignored. Is this a reasonable assumption? (I could not find any evidence for this). In addition, historical eutrophication in the Baltic may have caused significant storage of dissolved nutrients deep in the sediment (i.e. ammonia, phosphate, sulphide), which are not accounted for in the model. Can these be ignored – what is the effect of ignoring these on long-term simulations?

In addition, the dependencies of the biogeochemical processes on the model variables are so complex that it is very difficult to see how these processes are affected. For instance the formula (5), which essentially describes the dependency of denitrification on water-column oxygen and biota, has 4 "fitting" parameters – to what data have these

been fitted? The P-sequestration formula (formula 7) has even 8 "fitting" parameters. On line 187, it is said that it is difficult to constrain the new parameters. Does this mean that these parameters have not been fitted at all – and if they have, on which data? And why would instead running sensitivity analyses by changing the Ebio parameter be a valid alternative? A little more effort in showing that these dependencies are realistic is required. (and where is formula 6?).

I also find the lack of any comparison of model output with biogeochemical sediment data worrisome. On L 313, the authors claim that they cannot "properly validate the simulated sediment stocks or fluxes due to a lack of large–scale data and insufficient understanding of the multitude of mechanisms underlying the biogeochemical transformations and fluxes".  The first part (lack of data) does not do justice to the multiple biogeochemical studies in the Baltic that have recorded sediment-water exchange fluxes, and measured sediment concentration profiles in great detail. Also, I do not agree with the statement that there is "insufficient understanding" of biogeochemistry. As a quantitative science, biogeochemistry is at least as (and probably much more) advanced as biology!  And even if it were true that we do not understand the biogeochemistry, why would we then trust the simple parameterisations that are used in this manuscript?

In summary, as much as I like the conclusions from this paper, the authors need to try a bit harder to convince that biogeochemistry in the Baltic can be predicted based on presumed effects of biological activity on N and P removal.

We agree with the reviewer that organic matter processing in sediments has traditionally been studied in different fields of science with differing foci and assumptions. An excellent treatise of this subject can be found in the recent review by Middelburg (2018). As stated by Middelburg, we also believe that while there is merit in the traditional approaches, there is added benefit in interdisciplinary approaches bridging this gap. For example, most biogeochemical models of sediment diagenesis include the bioturbation of animals, but only represent their consumption of organic matter and secondary production implicitly in a bulk formulation. On the other hand, few biological models resolve the dynamic coupling between benthic animals and their sedimentary resources. In addition to Middelburg (2018), several other recent reviews and perspectives have called for interdisciplinary approaches merging the biological and biogeochemical as well as benthic and pelagic research traditions (Ehrnsten, 2020; Lessin et al., 2018; Snelgrove et al., 2014, 2018). We believe that our approach should be well suited for the current journal, as its aim is to "cover interactions between the biological, chemical, and physical processes". Or to cite the concluding remark of Middelburg (2018): "I hope that colleagues studying marine sediments are aware that "bio-" in sediment biogeochemistry is more than just microbiology".

We have added several justifications of our choice of approach and methodology to the introduction, as also requested by Reviewer 1. The new introduction is found on lines 26-69.

It is true that all models are simplified representations of reality, and the level of complexity and detail frames the questions that a model can answer reliably. Therefore, great care should be taken in choosing the appropriate model formulations based on the question(s) being asked. Here, our main focus is on the biogeochemistry of the Baltic Sea as a coupled benthic-pelagic system. This means that we are primarily focussing on basin-wide spatial and long temporal (days to decades) scales. Many physicalbiogeochemical ecosystem models working on similar scales choose to treat the sediments as a reactive boundary layer, where sinking organic matter is immediately transformed to inorganic compounds and returned to the pelagic. Soetaert, Middleburg, Herman & Buis (2000) reviewed and tested different approaches to couple benthic and pelagic biogeochemical models in coastal shelf systems (from no to vertically resolved sediment models), and concluded that the best choice is a vertically integrated dynamic sediment model of the type used in BALTSEM, because of an optimal balance between computational demand and accuracy attained in terms of e.g. mass budgeting and seasonality of benthic-pelagic solute fluxes. Two decades later, computational resources have increased, but we still argue that including a vertically resolved Reactive-Transport-Model (RTM) or similar for the sediments remains suboptimal. In the case of BALTSEM, each sediment variable is resolved at each depth meter in thirteen basins, amounting to 1349 individual RTMs that would need to be run in parallel. In addition to the increased running time, parameterisation would be difficult and time-consuming and the added complexity would reduce interpretability and traceability of results (Levins, 1966; Robinson, 2008). Some development in the integration of vertically resolved sediment and pelagic biogeochemistry is ongoing, but these models generally need to trade off complexity in other parts. For example, Radtke et al. (2019) implement such a model in one dimension at a few individual sites and omit dynamical modelling of the physics.

In the early days of BALTSEM development, Savchuk & Wulff (1996) developed and tested the use of a detailed process-oriented sediment model in BALTSEM with solids and solutes as separate state variables. However, they found that data to parameterize and verify the various processes on a system level were largely lacking. They also found that a simplified version with only one state variable for each sediment nutrient pool gave comparable results. In the approach used in BALTSEM since then, the solutes in the sediments are not prognostic state variables, hence, solutes produced or consumed in the sediments are directly causing exchange with the water column. We would argue that on a long-term ecosystem scale, the omission of short-term storage of solutes in pore waters does not significantly hamper model functionality. For further discussion and reasoning behind the formulations, we refer the reader to previous BALTSEM publications (Savchuk et al., 2012; Savchuk and Wulff, 1996, 2001) as well as other model descriptions using similar formulations (Capet et al., 2016; Isaev et al., 2020; Samuelsen et al., 2015).

Regarding the ability of BALTSEM to reproduce the long-term storage of nutrients in sediments in response to eutrophication, we would like to point to two previous studies, showing the simulated build-up of nutrient stores in sediments (Gustafsson et al., 2012) and the consequent increase in benthic fauna (Ehrnsten et al., 2020). While quantitative data to validate the sediment pools themselves is scarce, BALTSEM has been show to accurately to reproduce the long-term development of pelagic nutrient pools (Gustafsson et al., 2012; Savchuk et al., 2012). We consider this important indirect validation of sediment pools and dynamics, as it would not be possible to reproduce the time-lag in pelagic nutrient pools compared to inputs without a proper representation of sediment pools and processes.

The reviewer shows some concern regarding the complexity of sediment biogeochemical model formulation, in particular the equations for apportionment of mineralized N into NH, NO and $N_2$ ($v_{NOXY}$ and $\eta_N$) and mineralized P into release versus sequestration ($\eta_P$). These mathematical formulations are indeed not the most accessible, and we would therefore like to open them up with a graphical presentation of the shape of the curves in relation to bottom water oxygen concentration below. $v_{NOXY}$ gives the fraction of mineralized N released as NH in relation to oxygen concentration. $\eta_N$ ("eta$_N$") defines the fraction of oxidized N released as NO. The other fraction is denitrified. $\eta_P$ gives the fraction of mineralized P that is sequestered in the sediments. Here, the curve is shown for salinity > 5. The second term in $\eta_P$, $fsal$ is a step-like function which leads to higher P sequestration in the Bothnian Bay compared to all other basins. This is used a proxy for the higher availability of phosphate-binding agents in this basin.

[Figure]

The sigmoid form of the curves requires several parameters. We realize that calling these parameters "fitting constants" was misleading, as they are not independent parameters resulting from curve fitting to a specific dataset. Instead they are based on a general understanding of these processes put into mathematical terms (Savchuk and Wulff, 1996). We will therefore refer to them as just parameters in the future (lines 178, 183, 190, 827, 834, Table A2).

The addition of bioturbation enhancement to these formulations is also based on a qualitative understanding rather than fitting to data. As explained in the manuscript in more detail, we assume that bioturbation increases oxygen penetration into the sediments, thus shifting the curves in relation to bottom water oxygen concentration. As we do not know exactly how much the curves should be shifted, we performed a sensitivity analysis with a range of values for the parameter $E_{max}$ in the bioturbation enhancement formulation.

We realize that the statement on line 313 about lack of data was badly formulated and we have now removed it from the manuscript. There is indeed a wealth of research on sediment biogeochemistry in the Baltic Sea. What we meant to say is that a comprehensive compilation of data on sediment stocks and fluxes on the scale needed for model validation is missing. It would be beyond the scope of this study to compile such a validation dataset. When writing the original manuscript, we considered citing a range of benthic-pelagic fluxes measured in the Baltic as validation for our model estimates, but concluded that citing fluxes out of context does not do justice to the research performed and does not add much scientific value to the current study. For example, a recent compilation of sediment-water dissolved inorganic P fluxes measured in the Baltic Sea (Berezina et al., 2019) gives a range of ca -29 to +87 mg P $m^{-2}$ $d^{-1}$. Without a proper upscaling exercise accounting for the context of each study (which would be the subject of a comprehensive review), we do not believe that these numbers are of much value as validation.

Thus, beyond the validation of benthic fauna stocks and comparison to sediment C:N:P complied by Cederwall and Elmgren (2001), we refrain from formal validation of sediment stocks and fluxes. However, a qualitative, and partly quantitative, comparison of model results to the current understanding of sediment processes and the impact of benthic fauna on them is included in the discussion section 4.2. As stated above, we also believe that the validation of pelagic biogeochemistry (Appendix B) is an indirect validation for sediment stocks and fluxes, as the benthic and pelagic are strongly coupled in this system. We have also added a discussion on ways forward to improve model validation and reliability to the last chapter "5. Conclusion and outlook" (lines 450-468).

"Even though these large-scale simulations contain a large degree of uncertainty, they are an important complement to empirical studies, which for practical reasons can only consider temporally and spatially limited parts of the system (Boyd et al., 2018; Snelgrove et al., 2014). To improve the confidence in simulation results, we see two major ways forward. First, as all models contain different formulations, assumptions and uncertainties, implementing benthic fauna components in other physical-biogeochemical models and comparing the results would greatly increase the strength of evidence for those results where different models agree. This kind of ensemble modelling is increasingly used in climate change research, and has also been applied in the context of Baltic Sea biogeochemistry (Meier et al., 2012, 2018; Murphy et al., 2004). We hope that the publication of the benthic model formulations stimulates the development of benthic fauna modules in other models of the Baltic Sea ecosystem and beyond. Even though the current model implementation is only applicable to the brackish parts of the Baltic Sea due to a lack of functional groups present in the marine parts, the inclusion of additional functional groups using the existing groups as a template would be straightforward technically. The main challenges are the parameterisation of group-specific rates as well as managing the increased complexity.

Second, a comprehensive compilation of observational data on sediment stocks and fluxes would be needed for improved model validation. Such data is collected for monitoring and research purposes by a great number of institutions around the Baltic Sea, but a comprehensive, open-access, quality-controlled collection of this data is lacking. The Baltic Environment Database (BED) has been invaluable for both model development and validation of pelagic physics and chemistry. While this data can be used as indirect validation of benthic model processes in the strongly coupled system, we call for the development of a "Benthic BED" to facilitate future model development. A

comprehensive collection of observational data would also facilitate the identification of knowledge gaps and future research priorities."

**References in review response**

Berezina, N. A., Maximov, A. A. and Vladimirova, O. M.: Influence of benthic invertebrates on phosphorus flux at the sediment−water interface in the easternmost Baltic Sea, Mar. Ecol. Prog. Ser., 608, 33–43, doi:10.3354/meps12824, 2019.

Capet, A., Meysman, F. J. R., Akoumianaki, I. and Soetaert, K.: Integrating sediment biogeochemistry into 3D oceanic models: A study of benthic-pelagic coupling in the Black Sea, Ocean Model., 101, 83–100, doi:10.1016/j.ocemod.2016.03.006, 2016.

Ehrnsten, E.: Quantifying biomass and carbon processing of benthic fauna in a coastal sea – past, present and future, University of Helsinki., 2020.

Ehrnsten, E., Norkko, A., Müller-Karulis, B., Gustafsson, E. and Gustafsson, B. G.: The meagre future of benthic fauna in a coastal sea – benthic responses to recovery from eutrophication in a changing climate, Glob. Chang. Biol., 26, 2235–2250, doi:10.1111/gcb.15014, 2020.

Gustafsson, B. G., Schenk, F., Blenckner, T., Eilola, K., Meier, H. E. M., Müller-Karulis, B., Neumann, T., Ruoho-Airola, T., Savchuk, O. P. and Zorita, E.: Reconstructing the development of Baltic Sea eutrophication 1850-2006, Ambio, 41(6), 534–548, doi:10.1007/s13280-012-0318-x, 2012.

Isaev, A., Vladimirova, O., Eremina, T., Ryabchenko, V. and Savchuk, O.: Accounting for Dissolved Organic Nutrients in an SPBEM-2 Model: Validation and Verification, Water, 12(5), 1307, doi:10.3390/w12051307, 2020.

Lessin, G., Artioli, Y., Almroth-Rosell, E., Blackford, J. C., Dale, A. W., Glud, R. N., Middelburg, J. J., Pastres, R., Queirós, A. M., Rabouille, C., Regnier, P., Soetaert, K., Solidoro, C., Stephens, N. and Yakushev, E.: Modelling marine sediment biogeochemistry: Current knowledge gaps, challenges, and some methodological advice for advancement, Front. Mar. Sci., 5, 1–8, doi:10.3389/fmars.2018.00019, 2018.

Levins, R.: The strategy of model building in population biology, Am. Sci., 54(4), 421–431, 1966.

Middelburg, J. J.: Reviews and syntheses: to the bottom of carbon processing at the seafloor, Biogeosciences, 15(2), 413–427, doi:10.5194/bg-15-413-2018, 2018.

Radtke, H., Lipka, M., Bunke, D., Morys, C., Woelfel, J. and Cahill, B.: Ecological ReGional Ocean Model with vertically resolved sediments ( ERGOM SED 1 . 0 ): coupling benthic and pelagic biogeochemistry of the south-western Baltic Sea, , 275–320, 2019.

Robinson, S.: Conceptual modelling for simulation Part I: Definition and requirements, J. Oper. Res. Soc., 59(3), 278–290, doi:10.1057/palgrave.jors.2602368, 2008.

Samuelsen, A., Hansen, C. and Wehde, H.: Tuning and assessment of the HYCOM-NORWECOM V2.1 biogeochemical modeling system for the North Atlantic and Arctic oceans, Geosci. Model Dev., 8(7), 2187–2202, doi:10.5194/gmd-8-2187-2015, 2015.

Savchuk, O. and Wulff, F.: Biogeochemical Transformations of Nitrogen and Phosphorus in the Environment - Coupling Hydrodynamic and Biogeochemical Processes in Models for the Baltic Proper., 1996.

Savchuk, O. and Wulff, F.: A Model of the Biogeochemical Cycles of Nitrogen and Phosphorus in the Baltic, in A Systems Analysis of the Baltic Sea. Ecological Studies, vol 148, edited by F. W. Wulff, L. A. Rahm, and P. Larsson, pp. 373–415, Springer, Berlin Heidelberg., 2001.

Savchuk, O. P., Gustafsson, B. G. and Müller-Karulis, B.: BALTSEM - a marine model for desicion support within the Baltic Sea Region. BNI Technical report No 7., 2012.

Snelgrove, P. V. R., Thrush, S. F., Wall, D. H. and Norkko, A.: Real world biodiversity-ecosystem functioning: A seafloor perspective, Trends Ecol. Evol., 29(7), 398–405, doi:10.1016/j.tree.2014.05.002, 2014.

Snelgrove, P. V. R., Soetaert, K., Solan, M., Thrush, S., Wei, C.-L., Danovaro, R., Fulweiler, R. W.,

Kitazato, H., Ingole, B., Norkko, A., Parkes, R. J. and Volkenborn, N.: Global carbon cycling on a heterogeneous seafloor, Trends Ecol. Evol., 33(2), 96–105, doi:10.1016/j.tree.2017.11.004, 2018.

Soetaert, K., Middelburg, J. J., Herman, P. M. J. and Buis, K.: On the coupling of benthic and pelagic biogeochemical models, Earth-Science Rev., 51, 173–201, doi:10.1016/S0012-8252(00)00004-0, 2000.

---

## Author Response (AR2)

Dear Prof. Middelburg,

Thank you for your insightful evaluation of the revision process. While we agree with reviewer #1 that more data-model comparisons would strengthen the manuscript, we welcome your invitation to stay with the present version. As we explained in the earlier review responses, we do not see any realistic alternatives for including more comparisons due to a lack of observational data on an appropriate scale. Please find the final version of the manuscript re-uploaded in the submission system. No changes have been made compared to the previous version, except for the removal of two superfluous commas.

With best regards,
on behalf of all authors,

Eva Ehrnsten